# CHAIN-OF-GENERATION: PROGRESSIVE LATENT DIFFUSION FOR TEXT-GUIDED MOLECULAR DESIGN

## ABSTRACT

Text-conditioned molecular generation aims to translate natural-language descriptions into chemical structures, enabling scientists to specify functional groups, scaffolds, and physicochemical constraints without handcrafted rules. Diffusion-based models, particularly latent diffusion models (LDMs), have recently shown promise by performing stochastic search in a continuous latent space that compactly captures molecular semantics. Yet existing methods rely on one-shot conditioning, where the entire prompt is encoded once and applied throughout diffusion, making it hard to satisfy all the requirements in the prompt. We discuss three outstanding challenges of one-shot conditioning generation, including the poor interpretability of the generated components, the failure to generate all substructures, and the ambiguity in considering all requirements simultaneously. We then propose three principles to address those challenges, motivated by which we propose **Chain-of-Generation (CoG)**, a training-free multi-stage latent diffusion framework. CoG decomposes each prompt into curriculum-ordered semantic segments and progressively incorporates them as intermediate goals, guiding the denoising trajectory toward molecules that satisfy increasingly rich linguistic constraints. To reinforce semantic guidance, we further introduce a post-alignment learning phase that strengthens the correspondence between textual and molecular latent spaces. Extensive experiments on benchmark and real-world tasks demonstrate that CoG yields higher semantic alignment, diversity, and controllability than one-shot baselines, producing molecules that more faithfully reflect complex, compositional prompts while offering transparent insight into the generation process.

## 1 INTRODUCTION

Text-conditioned molecular generation is an emerging and promising research direction at the intersection of natural language processing and molecular design. This task aims to generate molecular structures that match a given textual description, involve scientists directly in the design process, and enable intuitive and flexible control over molecular properties. This approach utilizes natural language as input, allowing users to specify high-level goals, such as desired functional groups, scaffolds, or physicochemical properties, thereby eliminating the need for expert-crafted constraints or domain-specific encoding schemes. Early work in this field primarily focused on retrieval-based approaches (Edwards et al., 2021; Zeng et al., 2022), followed by direct generation methods such as text-to-SMILES sequence translation (Edwards et al., 2022), where molecules are represented using the Simplified Molecular Input Line Entry System (SMILES) (Weininger, 1988). These approaches commonly employ a sequence-to-sequence modeling paradigm, directly mapping natural language prompts to SMILES strings.

Recent years have witnessed the impressive imaging generation capability of diffusion-based models (Ho et al., 2020), which model data through iterative denoising steps. This success has inspired their application to molecular generation tasks (Gong et al., 2024; Chang & Ye, 2024), where their stochastic sampling process facilitates the creation of structurally diverse and novel molecules. Among these, latent diffusion models (LDMs) (Rombach et al., 2022; Zhu et al., 2024) stand out for their efficiency and scalability. Instead of generating molecules directly in molecule space, LDMs operate over a learned continuous latent space, where generative modeling becomes more tractable, and the structural semantics of molecules can be more compactly captured. This decoupling of representation and generation allows LDMs to produce chemically valid and diverse molecules while

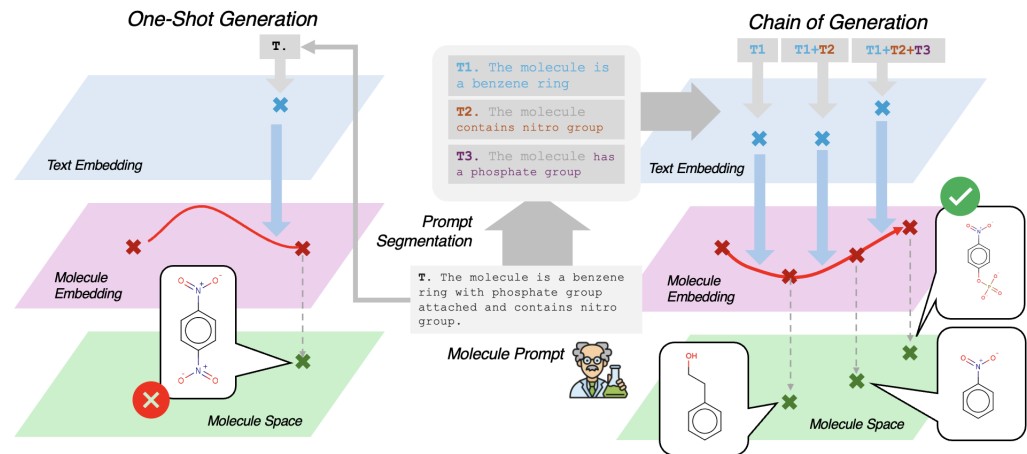

Figure 1: Overview of the proposed chain-of-generation (CoG). **Left:** Existing diffusion approaches (e.g., (Zhu et al., 2024)) where a text prompt $T$ is projected to a vector in the text-embedding space used for guiding the diffusion in the molecule embedding space for generation. **Right:** The proposed CoG approach firstly segments the text prompt into a set of components $(T_1, T_2, T_3)$ and progressively includes the components into smaller prompts to guide multi-staged diffusion for molecule generation.

remaining computationally efficient. LDMs typically follow a two-stage framework: (1) learning bidirectional projections that map molecular structures into and out of a rich latent space that captures molecular information; (2) guiding a generative search process within this latent space using natural language signals, then decoding the resulting latent vectors back into valid molecules.

While recent works focus on the first stage by improving molecular encoding and pretraining strategies (Chang & Ye, 2024), the second stage remains underdeveloped. In particular, without robust mechanisms to effectively relate the textual prompt space with the molecular latent space, the model may fail to identify the correct generation trajectory. This weak cross-modal correspondence can lead to semantically misaligned or structurally incoherent molecules, limiting both generation quality and controllability. Additionally, evaluating the semantic quality of generated molecules remains a critical open challenge. Existing evaluation metrics are mostly adapted from NLP, such as BLEU (Papineni et al., 2002) or Levenshtein distance (Lcvenshtcin, 1966), and do not fully capture whether a molecule truly reflects the intended semantics of a prompt, especially at the graph level. This makes human-in-the-loop assessment increasingly important to understand model behavior. Finally, most existing methods, such as 3M-Diffusion (Zhu et al., 2024), rely on a one-shot conditioning paradigm, where the entire textual prompt is encoded once and used to guide the generation process. However, as shown in Fig. 1, we observe that this approach often fails when prompts are semantically complex or composed of multiple components. There are three main challenges of one-shot conditioning that limit the capacity of generation. First, as one-shot conditioning only uses the prompt once at the beginning of the generation, it is hard to attribute the generated components back to the prompt. Second, when there are multiple requirements in the prompt, the generated molecules may omit key substructures or components. Third, one-shot conditioning takes all the requirements at once, making it challenging to resolve and plan the generation procedure.

We then propose three principles to address these challenges. First, the generation process should be *guided* rather than end-to-end. Second, early-stage prompts should be utilized as a *regularization* in the later stage of generation. Third, the generation should consider *granularity*, ranging from coarse structures to fine-grained components. Building on this idea, we introduce a *training-free* Chain-of-Generation (CoG) framework for molecule generation. To further enhance semantic guidance, we introduce a post-alignment learning stage that strengthens the correspondence between textual signals and molecular latents. Finally, we perform extensive experiments on both synthetic and real-world datasets, showing that CoG yields more faithful, interpretable, and controllable generation.

## 2 RELATED WORK

Text-conditioned molecular generation is an emerging task at the intersection of computational chemistry and multimodal learning. It aims to generate molecular structures from natural language prompts, enabling intuitive, goal-directed molecular design. Early works approached this as a *retrieval* problem, learning joint embeddings to align text and molecular representations (Edwards et al., 2021; Zeng et al., 2022). While effective, these methods are constrained to pre-existing molecules and cannot generate novel structures. To overcome this, generation-based models emerged, typically translating text to SMILES strings (Weininger, 1988). Models such as MolT5 (Edwards et al., 2022) and ChemT5 (Christofidellis et al., 2023) use encoder–decoder transformers to map prompts to molecules, benefiting from large pretrained language models. However, prior sequence-to-sequence approaches exhibit severe *mode collapse*: despite the large solution space implied by flexible prompts, they repeatedly converge to a single rigid output. A second limitation is *chemical invalidity*. Because SMILES impose an artificially defined string grammar, models must implicitly relearn these rules, often producing syntactically or chemically invalid molecules. An illustrative example is provided in Appendix E. Graph-based diffusion methods alleviate the issue of invalid SMILES generation by operating directly on molecular graphs. However, performing diffusion in the original discrete graph space remains problematic: permutation invariance introduces many isomorphic orderings, and training often relies on auxiliary reconstruction losses (Vignac et al., 2022; Simonovsky & Komodakis, 2018). These losses are poorly suited for molecular graphs, where even small inconsistencies can distort chemical properties. To avoid these limitations, LDMs (Zhu et al., 2024) encode graphs into a continuous latent space and perform text-conditioned diffusion there, preserving validity while enabling stochastic, conditional exploration for more diverse and novel molecules. Our work builds on LDMs, with a specific focus on improving the language-conditioned searching process. Rather than modifying molecular representations or encoders, we propose a multi-step planning framework that decomposes complex prompts into semantic subgoals, guiding the diffusion process to search within an aligned subspace in a more faithful and interpretable manner.

## 3 PRELIMINARY AND MOTIVATION

### 3.1 PRELIMINARY

LDMs first learn a bidirectional mapping between molecular graphs and a continuous latent space, and then apply a conditional diffusion process to perform guided sampling in this space using natural language prompts, followed by decoding the resulting latent vectors back into molecular structures (Rombach et al., 2022). Zhu et al. (2024) achieve this by training a variational graph auto-encoder (Kingma et al., 2013), comprising a graph encoder and a graph decoder. The encoder maps input molecular graphs into latent Gaussian distributions using a pre-aligned Graph Isomorphism Network (GIN) (Xu et al., 2018). The decoder reconstructs molecular structures from the latent vectors by a hierarchical variational autoencoder (HierVAE) (Jin et al., 2020). The model is trained by minimizing the Evidence Lower Bound (ELBO):

$$\mathcal{L}_{\text{elbo}} = -\mathbb{E}_{q(g|G)}[\log p(G \mid \hat{G})] + D_{\text{KL}}\left[q(g \mid G) \,\|\, p(g)\right],\tag{1}$$

where $\hat{G} = D(g)$ denotes the reconstructed molecular graph generated by the decoder $D$. The latent representation $g$ is sampled from the approximate posterior $q(g \mid G)$, where $G$ is the input molecular graph. Specifically, the encoder first processes $G$ using a GIN, and then two separate multilayer perceptrons (MLPs) are applied to predict the mean and standard deviation of the latent Gaussian distribution. Using the reparameterization trick (Kingma et al., 2013), a continuous latent vector $g$ is sampled from this distribution. The Kullback–Leibler divergence term encourages the learned posterior $q(g \mid G)$ to stay close to the prior $p(g) = \mathcal{N}(0, I)$, thereby regularizing the latent space.

The text-conditioned generation is achieved by latent diffusion in the learned latent space. Starting from a clean latent representation $g_0$, Gaussian noise is added according to a predefined schedule to obtain a noisy latent vector: $z_t = \sqrt{\bar{\alpha}_t} g_0 + \sqrt{1 - \bar{\alpha}_t}\, \epsilon$, where $\bar{\alpha}_t$ denotes the noise level at timestep $t$, and $\epsilon \sim \mathcal{N}(0, I)$. A denoising network $\theta$ is then trained to recover $g_0$ from $z_t$, conditioned on the text embedding $c$, by minimizing the following:

$$\mathbb{E}_{t,c,g_0,\epsilon}\left[\left\|\epsilon - \theta\left(\sqrt{\bar{\alpha}_t} g_0 + \sqrt{1 - \bar{\alpha}_t}\,\epsilon,\; t,\; c\right)\right\|^2\right],$$

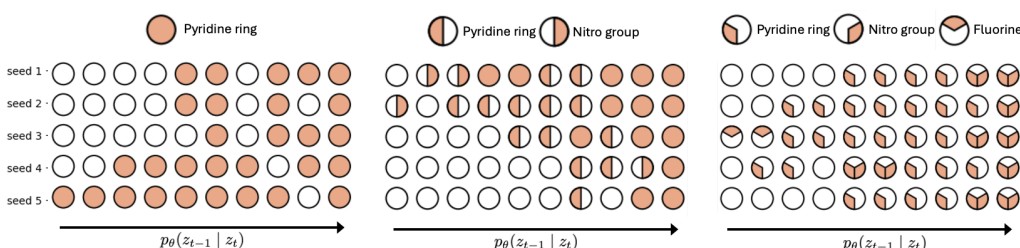

Figure 2: Generated molecules for three textual prompts. **Left**: "`The molecule is made of a pyridine ring.`" **Middle**: "`The molecule is a pyridine ring with a nitro substituent.`" **Right**: "`The molecule is made of a pyridine ring substituted with both a nitro group and fluorine atoms.`" Note that as the complexity of the prompt increases, one-shot conditioning may not be able to satisfy all the requirements. Moreover, latent diffusion ensembles each component progressively during generation.

which learns a probabilistic mapping from noisy latent representations to clean molecular embeddings, enabling the search for target molecules through the latent space based on natural language signals.

## 3.2 MOTIVATION

Recent advances in latent diffusion models have shown promise in text-conditioned molecular generation, but complex prompts often remain challenging with one-shot conditioning: models sometimes fail to construct molecular structures that fully reflect all required semantics. To probe this issue, we investigate how diffusion models, as black-box generators, parse natural language descriptions and incrementally assemble molecular structures during denoising. We conduct a trajectory analysis by decoding partially denoised latents and visualizing intermediate outputs, enabling us to examine how structural features emerge and evolve throughout the generation process.

We begin with simple prompts (e.g., "`This molecule is made of a pyridine ring`") and gradually increase complexity by adding compositional elements. We see that when the prompt includes three or more structural keywords, model performance begins to diverge significantly. Fig. 2 shows an example. The prompt is constructed from compositional vocabulary in our training set. In the visualization, colored regions correspond to substructures correctly generated in accordance with the prompt, while uncolored parts indicate incomplete alignment. Notably, as we increase the number of components in the prompt, latent diffusion cannot simultaneously handle all the requirements, indicating the limitation of one-shot conditioning. Additionally, the model tends to resolve larger, high-level scaffolds first, for example, the pyridine ring, which is designated as the core structure in the prompt. Then, as denoising proceeds, it decodes peripheral substructures piece by piece (e.g., attaching a nitro group) that serve as modifiers or substituents. This illustrates that the model progressively integrates semantic elements from the prompt, aligning more closely with the intended meaning over the course of denoising, a behavior rarely observed in sequence-to-sequence models. Motivated by this observation, we propose our Chain-of-Generation framework in Section 4.3.

## 4 MOLECULE CHAIN-OF-GENERATION

Motivated by the aforementioned observation of one-shot conditioning in Section 3.2, we propose a novel **Chain-of-Generation (CoG)** framework in this section, which decomposes each prompt into semantically meaningful segments and progressively guides the generation through these stages. Importantly, CoG operates entirely at sampling time without requiring retraining. We first discuss three challenges of one-shot conditioning. We then propose three principles of latent diffusion to address those challenges. Equipped with that understanding, we introduce our CoG framework. Finally, a post-alignment strategy is proposed to enhance the generation.

### 4.1 CHALLENGE OF ONE-SHOT CONDITIONING

Although recent LDMs have demonstrated promising capabilities in text-conditioned molecular generation, a key challenge remains in how language guidance is incorporated into the latent gener-

ation process. Existing methods encode the entire textual prompt once and use it to condition the entire generation process. However, this one-shot conditioning strategy struggles with complex or compositional prompts. In Section 3.2, empirical results are shown to illustrate the failure of one-shot conditioning. In this section, we further analyze the challenges of one-shot conditioning from three perspectives. **C1: Poor interpretability**. Since the prompt is used only once at the beginning, it is difficult to attribute parts of the generated molecule to specific components of the text, limiting transparency and controllability. **C2: Missing substructures**. When prompts describe multiple functional groups or hierarchical structures, the model often fails to assemble all parts correctly, omitting key molecular subcomponents. As in Fig. 2, a prompt specifying (1) a pyridine ring, (2) a nitro group, and (3) fluorine yields a generated molecule that includes only the first two, omitting the fluorine atoms with one-shot conditioning. **C3: Ambiguous conditioning**. One-shot strategies treat all prompt information simultaneously throughout the generation, ignoring the relationships among components and making it difficult to position components correctly within the overall structure. To address these limitations, we propose three principles of latent diffusion in the next section.

## 4.2 THE ART OF LATENT DIFFUSION

In this section, we propose three principles to address the above three challenges, respectively.

*Guided Search Process.* To address **C1**, each attribute of the molecule is supposed to be traced back to the corresponding component in the prompt. Thus, rather than jumping blindly from Gaussian noise to the final manifold, the latent walks along a continuous path of feasible solutions where each stage acts as a bridging distribution that keeps the sample within a high-density overlapping region.

*Early-Stage Prompts as Regularization.* To help eliminate missing substructures in generation in **C2**, it is critical to consider both the current focus and historical information properly. The generation process should preserve the existing structure by leveraging historical information, while accurately producing the next components in accordance with the current stage.

*Curriculum-Style Generation.* **C3** highlights the difficulty of focusing on all the prompt components with one-shot conditioning. An alternative solution is to adopt a coarse-to-fine generation schedule. In generation, early stages establish the global structure, and later stages add details on top of it. Viewed heuristically, this resembles a continuation/curriculum procedure that makes small updates around the previous solution, which we find to stabilize sampling and improve adherence to later prompt components without overwriting earlier ones.

## 4.3 CoG: THE WISDOM OF DECOMPOSING AND COMPOSING

Chain-of-Thought (CoT) reasoning in LLMs has shown how complex problems can be solved more reliably by decomposing them into sequential sub-tasks rather than tackling them all at once (Wei et al., 2022; Yao et al., 2023). Inspired by this philosophy, we design Chain-of-Generation (CoG) for molecular diffusion models. Unlike CoT, which operates in natural-language reasoning, CoG introduces a novel curriculum within the latent diffusion process. It begins by *decomposing* a molecular prompt into semantically meaningful components that reflect chemical granularity—such as scaffolds, functional groups, and fine-grained modifiers, and then organizes these components into a coarse-to-fine hierarchy. Each stage of diffusion is explicitly conditioned on the cumulative sub-prompts, allowing the model to progressively *assemble* molecular structures while preserving previously generated substructures. This structured planning and progressive denoising transform CoG into more than a direct adaptation of CoT: it is a chemically grounded mechanism that enables diffusion to generate molecules with higher semantic fidelity, interpretability, and controllability.

**Prompt Decomposition and Planning.** The first step of CoG is to decompose a text prompt $T$ into a set of $N$ prompt segments $\mathcal{S}(T) = [T_1, T_2, \ldots, T_N]$. A planning process $\mathcal{P}$ then converts the segments into a set of sub-prompts:

$$\mathcal{P} \odot \mathcal{S}(T) = [T_1, T_1 + T_2, \ldots, T_1 + T_2 + \cdots + T_N \equiv T], \qquad (2)$$

which describes a sequence of sub-goals that align with our hierarchical planning objective. In the illustrating example in Fig. 1, $T$ is `"The molecule is a benzene ring with a phosphate group attached and contains a nitro group."` We decompose it

into: $T_1$: "The molecule is a benzene ring.", $T_2$: "The molecule contains a nitro group.", and $T_3$: "The molecule has a phosphate group attached."

Researchers with a chemical background can segment the prompts manually, but this can be inefficient given a massive generation task. We propose an automated *hierarchical planning strategy* for prompt segmentation using a large-language model, inspired by how chemists typically approach molecule construction: The first sub-prompt $T_1$ is expected to capture the molecule's primary scaffold or core structure that defines the molecular identity. Subsequent sub-prompts $T_2, T_3, \ldots$ incrementally introduce additional structural details, such as functional groups (e.g., nitro, methyl, hydroxyl) that attach to the core. In later stages, the prompts typically encode finer-grained constraints, including specific atom types, substitution positions, or stereochemistry (e.g., "a chlorine atom at position 3" or "meta-substituted amine"). This planning strategy enables the model to prioritize the generation of key substructures early on and refine them step-by-step in a chemically interpretable and semantically grounded manner. Our strategy allows the model to construct the most chemically meaningful substructures early in the diffusion process, leading to more stable and interpretable intermediate representations. It promotes structural consistency by grounding the initial steps on high-confidence semantic anchors. In Appendix B, we provide the instruction prompt we provided to LLM for segmentation and compare different LLM planning performances.

**Progressive Latent Diffusion.**   In Eq. (2), we obtain a sequence of sub-prompts that incrementally refine the semantic constraints guiding generation while maintaining the inherited parts in the text space. At each stage $k$, we condition the model on $T_1 + \cdots + T_k$ and use the latent vector generated in the previous stage $z_{k-1}$ as the initialization for the next denoising process. Formally, we iterate:

$$z_k \sim \text{Denoising}(\text{init} = z_{k-1}, \text{ cond} = E_t(T_{1:k})).$$

This chain-of-generation procedure enables the model to more faithfully follow the semantic progression of complex prompts. Earlier steps in the chain establish the coarse molecular backbone, while later stages introduce finer-grained modifications, such as specific functional groups or spatial arrangements. Importantly, the model should exhibit high confidence in retaining structures from previous steps. To achieve this, during later stages of the chain, we apply only the partial denoising trajectory determined by a step hyperparameter $s_{\text{start}} \in [0, S]$. In practice, we start the stochastic denoising process from an intermediate noise level, typically within the last 20% of the diffusion steps. This ensures that critical elements from $z_{k-1}$ are largely preserved, while still allowing for prompt-conditioned stochastic exploration guided by $T_k$.

## 4.4 POST-ALIGNMENT

The central representation for diffusion-based generation in our framework is the resampled latent vector $g$, obtained via the reparameterization trick in the VAE. This latent vector serves as the reconstruction target during diffusion training and defines the subspace where molecules are generated. To ensure that generation is effectively guided by language, it is crucial that $g$ remains well aligned with the corresponding text embedding. Such cross-modal alignment enhances the semantic correspondence between molecular graphs and their textual descriptions, allowing the diffusion process to utilize prompt signals more effectively. In particular, by aligning $g$ with the text representation, the model can perform more faithful and controllable search in the molecular embedding subspace, progressively assembling structures that match the intended semantics.

To achieve this, we introduce an additional post-alignment stage, where contrastive learning is performed directly between the resampled latent $g$ and the text embedding extracted from SciBERT (Beltagy et al., 2019). During this stage, the graph encoder is frozen to fix the latent representation $g$, which serves as input to the diffusion model from Eq. (1). We conduct a contrastive learning *after* the latent vector $g$ is sampled, optimizing the following:

$$\mathcal{L}_{\text{con}} = -\frac{1}{|\mathcal{B}|} \sum_{(G_i, T_i) \in \mathcal{B}} \log \frac{\exp\left(\cos(\mathbf{g}_i, \mathbf{c}_i)/\tau\right)}{\sum_{j=1}^{|\mathcal{B}|} \exp\left(\cos(\mathbf{g}_i, \mathbf{c}_j)/\tau\right)},$$

where $\mathbf{g}_i = E_g(G_i)$ and $\mathbf{c}_i = E_t(T_i)$ denote the representations of the molecular graph $G_i$ and its corresponding text prompt $T_i$, $\mathcal{B}$ represents a batch of molecule–text pairs sampled from the training set, $\cos(\cdot)$ denotes cosine similarity, and temperature $\tau$ controls the sharpness of the distribution.

| Text Description | Ground Truth | Distance Metrics | Output |
|---|---|---|---|
| The molecule is a stilbenoid that is trans-resveratrol substituted at position 3 by a beta-D-glucosyl residue… It is a stilbenoid, a polyphenol, a beta-D-glucoside and a monosaccharide derivative… 

 C1=CC(=CC=C1/C=C/C2=CC(=CC(=C2)O[C@H]3[C@@H]([C@H]([C@@H]([C@H](O3)CO)O)O)O)O | | MACCS SIM 1.0 

 Levenshtein distance: 61 | 
 OCC1OC(Oc2cc(O)cc(C=Cc3ccc(O)cc3)c2)C(O)C(O)C1O |
| The molecule is an optically active form of [2,8-bis(trifluoromethyl)quinolin-4-yl]-(2-piperidyl)methanol having (+)-(11R,2'S)-erythro-configuration… It is an enantiomer of a (-)-(11S,2'R)-erythro-mefloquine. 

 C1CCN[C@@H]([C1)[C@@H]([C2=CC(=NC3=C2C=CC=C3C(F)(F)F)C(F)(F)F)O | | MACCS SIM 1.0 

 Levenshtein distance: 49 | 
 OC(c1cc(C(F)(F)F)nc2c(C(F)(F)F)cccc12)C1CCCCN1 |

Figure 3: Two examples where nearly identical molecular graphs are represented by drastically different SMILES strings. In such cases, NLP-based string metrics (e.g., Levenshtein) often assign a high distance score, failing to reflect the true structural correspondence. In contrast, our graph-based evaluation correctly captures scaffold and functional group similarity.

By applying the post-alignment strategy, we establish a robust latent diffusion backbone, denoted as **GraphLDM**. This backbone serves as the foundation for all subsequent experiments. In Sections 5.2 and 5.3, we show that augmenting GraphLDM with our proposed Chain-of-Generation strategy further improves semantic alignment and enhances controllability throughout the generation process.

## 5 EXPERIMENTS AND RESULTS

We conduct extensive experiments to evaluate the proposed framework. In Section 5.1, we first examine the shortcomings of existing NLP-oriented metrics for text-to-molecule generation and motivate our adoption of MACCS fingerprints together with more chemically faithful evaluation measures. In Section 5.2, we then assess performance on benchmark datasets, showing that **GraphLDM + CoG** consistently outperforms both SMILES-based methods and prior graph-based baselines. Finally, in Section 5.3, we analyze the role of prompt planning, comparing different large language models (LLMs) for text segmentation and interpretation, and further study curriculum versus anti-curriculum strategies as well as the impact of prompt regularization in a synthetic setting.

### 5.1 METRICS

Our evaluation relies on graph similarity computed from MACCS fingerprints using the RDKit (Bento et al., 2020). The MACCS fingerprint is a 166-bit, predefined keyset where each bit indicates the presence or absence of a specific substructure pattern, such as carbonyls or ring systems. The public 166-key subset is widely implemented in RDKit and it is commonly used throughout drug discovery (Durant et al., 2002; O'Hagan & Kell, 2015). Because these bits are computed from the *molecular graph* instead of the SMILES string, MACCS is naturally invariant to SMILES enumeration and canonicalization, making it representation-agnostic for our text-to-graph task.

Previous sequence-based works have often adopted evaluation metrics such as BLEU, Fréchet ChemNet Distance (FCD) (Brown et al., 2019), and Text2Mol (Edwards et al., 2021). However, many of these metrics are either NLP-oriented or designed for tasks with different objectives, making them unsuitable for our text-to-graph molecular generation setting. They reward token-level alignment and are highly sensitive to SMILES permutations even when the underlying graphs are identical. Text2Mol measures text–SMILES alignment for *retrieval* rather than pairwise, graph-level generation fidelity; and FCD compares *set-level* unconditional distributions learning instead of one-to-one prompt-molecule pairs. For these reasons, MACCS plus Tanimoto similarity provides a chemically meaningful, permutation-invariant basis for assessing performance in our setting. As illustrated

Table 1: Performance comparison on ChEBI-20 and PubChem datasets. All values are reported in percentages.

| Method | ChEBI-20 | | | | PubChem | | | |
|---|---|---|---|---|---|---|---|---|
| | BQI | Q-Cov | Q-Nov | Validity | BQI | Q-Cov | Q-Nov | Validity |
| **SMILES-based methods** | | | | | | | | |
| MolT5 small | 51.45 | 24.25 | 4.96 | 78.45 | 44.23 | 10.88 | 1.04 | 72.93 |
| MolT5 large | 55.95 | 20.35 | 2.23 | 98.08 | 51.44 | 16.27 | 1.45 | 94.15 |
| ChemT5 small | 57.6 | 23.88 | 3.25 | 96.99 | 52.69 | 20.05 | 2.47 | 91.91 |
| ChemT5 base | 57.87 | 25.08 | 3.45 | 97.25 | 52.97 | 20.77 | 2.75 | 90.65 |
| BioT5 base | 59.11 | 20.88 | 4.57 | 99.98 | 54.74 | 16.78 | 2.89 | **100** |
| BioT5 plus | 56.76 | 15.44 | 2.47 | **100** | 55.05 | 17.11 | 3.08 | **100** |
| **Graph-based methods** | | | | | | | | |
| 3M diffusion | 56.3 | 40.32 | 9.88 | **100** | 55.29 | 40.07 | 10.78 | **100** |
| GraphLDM | 59.43 | 43.05 | 11.02 | **100** | 57.79 | 40.88 | **12.56** | **100** |
| GraphLDM + CoG | **60.49** | **45.63** | **11.1** | **100** | **58.17** | **43.36** | 11.95 | **100** |

Table 2: Comparison of prompting strategies across different LLMs. All values are reported in percentages. The *validity* metric is excluded, as perfect chemical validity (100%) is guaranteed by our motif-based graph decoder for all combinations of LLM and prompting strategies.

| LLM | Prompting Strategy | BQI | Q-Cov | Q-Nov |
|---|---|---|---|---|
| | One-shot ($T$) | 49.6 | 31.69 | 10.03 |
| **GPT-4o** | One-shot ($T_s$) + CoG ($T_{s:m}$) + CoG ($T$) | 46.59 | 28 | 7.64 |
| | One-shot ($T_l$) + CoG ($T_{l:m}$) + CoG ($T$) | **48.46** | **31.07** | **8.42** |
| | One-shot ($T_l$) + CoG ($T_m$) + CoG ($T_s$) | 38.36 | 10.77 | 3.27 |
| **Gemini 2.5 Pro** | One-shot ($T_s$) + CoG ($T_{s:m}$) + CoG ($T$) | 45.59 | 25.85 | 7.13 |
| | One-shot ($T_l$) + CoG ($T_{l:m}$) + CoG ($T$) | **50.64** | **35.38** | *10.16* |
| | One-shot ($T_l$) + CoG ($T_m$) + CoG ($T_s$) | 34.32 | 4.62 | 1.22 |
| **Grok v3** | One-shot ($T_s$) + CoG ($T_{s:m}$) + CoG ($T$) | 47.02 | 28.92 | 7.85 |
| | One-shot ($T_l$) + CoG ($T_{l:m}$) + CoG ($T$) | *50.91* | *36.62* | **10** |
| | One-shot ($T_l$) + CoG ($T_m$) + CoG ($T_s$) | 35.29 | 6.15 | 1.78 |

in Fig. 3, two molecules with nearly identical graphs (scaffold and functional groups) can have drastically different SMILES strings. String-based metrics (BLEU/Exact/Levenshtein) therefore underrate the match, whereas MACCS Tanimoto correctly reflects the high structural similarity.

We report four evaluation metrics to assess model performance based on the MACCS. (1) **Validity** measures the proportion of chemically valid molecules among all generated outputs, ensuring that the generation process respects fundamental chemical rules. (2) **Qualified Novelty(Q-Nov)** evaluates the extent to which generated molecules, once they satisfy the textual prompt, are further novel and diverse, highlighting the model's ability to avoid redundancy and capture a broad range of structural patterns. (3) **Qualified Coverage(Q-Cov)** reflects how well molecules that satisfy the textual prompt explore new chemical space relative to known valid structures, highlighting the model's potential to generate practically useful candidates beyond the training distribution. (4) **Balanced Quality Index(BQI)** integrates multiple perspectives into a single score, providing a holistic view of generation quality by balancing both the fidelity and the creativity of the generation. The detailed formulations are provided in Appendix C.

## 5.2 BENCHMARK EVALUATION

We evaluate the effectiveness of our Chain-of-Generation strategy on benchmark datasets. Specifically, we conduct experiments on PubChem (Liu et al., 2023) and ChEBI-20 (Edwards et al., 2021), restricting all training, validation, and test molecules to fewer than 30 atoms to control molecular complexity. Following the progressive diffusion strategy described in Section 4.3, we divide each textual prompt into three stages: core structure, functional groups and modifiers, and stereochemical configurations. We employ Grok v3 for segmentation and apply our CoG strategy for curriculum-based prompting.

Table 1 reports the performance comparison on ChEBI-20 and PubChem. We denote our improved diffusion backbone as GraphLDM, which integrates post-alignment strategy to establish a strong starting point. Building on this, we apply the CoG for progressive conditional generation, which incorporates our proposed segmentation and curriculum prompting strategy. As shown in the table, GraphLDM + CoG achieves consistent improvements over the baselines (Pei et al., 2023; Christofi-dellis et al., 2023; Edwards et al., 2022; Zhu et al., 2024) across both datasets, particularly in BQI and Q-Cov, while maintaining perfect validity. These results highlight that CoG brings measurable benefits beyond baseline refinements, strengthening semantic alignment throughout the generation process. While Q-Nov occasionally shows a slight decrease, this is largely due to reduced variability among generated molecules—an expected trade-off for improved determinism, as CoG more reliably parses structural cues from the text. A detailed analysis by prompt segmentation is provided in Appendix D.

## 5.3 Prompt Planning and Ablations

To evaluate the effectiveness of prompt segmentation, planning strategy, and prompt regularization on our proposed Chain-of-Generation. We construct a synthetic dataset with controlled semantic structure consisting of zero-shot molecule–prompt pairs. Each prompt $T$ in this dataset is a compositional description of a molecular structure, designed such that it can be decomposed into three sequential subgoals: $T_1$, $T_2$, and $T_3$. To test whether different LLMs can understand and segment prompts according to our proposed coarse-to-fine planning strategy, we compare three representative models: GPT-4o (Hurst et al., 2024), Gemini 2.5 Pro (Google DeepMind, 2024), and Grok v3 (xAI, 2024). We define three levels of prompt components. $T_l$: "Large" components representing the primary scaffold of the molecule; $T_m$: "Medium" components representing functional groups or substituents; $T_s$: "Small" components containing fine-grained modifications such as specific atoms. We also evaluate three prompting strategies to guide the generation process: one-shot, fine-to-coarse, and coarse-to-fine.

From Table 2, we see that coarse-to-fine prompting consistently outperforms the one-shot baseline, whereas the fine-to-coarse strategy performs worse across all settings. This shows the importance of planning design in the CoG framework. When the model is forced to focus prematurely on atomic-level details, it later struggles to assemble a coherent molecule around a stable scaffold. By contrast, starting from the core structure and progressively introducing functional groups and modifications allows the model to anchor on the main skeleton and refine the molecule step by step, closely mirroring how human chemists approach molecular design.

We further conduct an ablation study to examine the effect of early-stage prompt regularization, as discussed in Section 4.2. Following the coarse-to-fine prompting strategy, for example, we generate an initial molecule from the one-shot prompt $T_l$, and then continue the chain-of-generation using only $T_m$ instead of the combined prompt $T_{l:m}$. The results show that when guided solely by a new intermediate prompt, the model tends to forget the previously established structure, even under structured planning. This leads to degraded final performance compared to the one-shot baseline.

Turning to prompt segmentation quality across different LLMs, we find that Grok v3 generally provides the most reliable results, correctly identifying the primary scaffold, functional groups, and fine-grained details in complex prompts. In contrast, GPT-4o occasionally mixes core and peripheral elements, sometimes overemphasizing atomic-level information too early or revisiting the scaffold at inappropriate stages (see Appendix A for comparison).

## 6 Conclusion

We proposed CHAIN-OF-GENERATION, a novel framework that brings structured planning into text-to-molecule synthesis. By decomposing prompts into semantically meaningful stages and guiding generation through progressive latent diffusion, CoG transforms one-shot generation into a transparent and controllable process. This design enhances semantic fidelity and substructure alignment, providing interpretability by linking prompt components to molecular features. Coupled with a post-alignment backbone, CoG demonstrates significant gains over existing baselines and introduces a principled approach that can inspire more interpretable and controllable molecular generation in future work.

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

# A    VISUALIZATION AND SEGMENTATION ACROSS LLMS

As illustrated in Fig. 4, GPT-4o occasionally fails to follow the correct order, resulting in the CoG process prematurely focusing on small decorative components during the early coarse stage, and subsequently struggling to reorganize the intermediate results to accommodate the core scaffold in the finer stages. In contrast, Grok v3 shows a more stable and semantically aligned understanding of hierarchical prompts.

| Primary structure? | GPT 4o | Gemini 2.5 Pro | Grok v3 |
|---|---|---|---|
| | P | | |
| | I | | |
| | F | F | |

Figure 4: Segmentation results across different LLMs

Table 3: Comparison of system prompt designs across different LLMs. All values are reported in percentages. The **validity** metric is excluded, as perfect chemical validity is guaranteed by our motif-based graph decoder.

| LLM | Prompting Strategy | BQI | Q-Cov | Q-Nov |
|---|---|---|---|---|
| | One-shot ($T$) | 49.6 | 31.69 | 10.03 |
| **One piece** | GPT-4o One-shot ($T_l$) | 36.20 | 9.23 | 2.43 |
| | GPT-4o One-shot$^-$ ($T_l$) | 35.96 | 9.23 | 2.34 |
| | Gemini 2.5 Pro One-shot ($T_l$) | 37.61 | 12.31 | 3.18 |
| | Gemini 2.5 Pro One-shot$^-$ ($T_l$) | 36.20 | 9.23 | 2.43 |
| | Grok v3 One-shot ($T_l$) | 37.84 | 12.31 | 3.29 |
| | Grok v3 One-shot$^-$ ($T_l$) | 39.83 | 15.38 | 4.39 |
| **Two pieces** | GPT-4o One-shot ($T_l$) + CoG ($T_{l:m}$) | 44.87 | 24.62 | 6.68 |
| | GPT-4o One-shot$^-$ ($T_l$) + CoG$^-$ ($T_{l:m}$) | 43.88 | 23.38 | 6.37 |
| | Gemini 2.5 Pro One-shot ($T_l$) + CoG ($T_{l:m}$) | 46.82 | 30.15 | 8.13 |
| | Gemini 2.5 Pro One-shot$^-$ ($T_l$) + CoG$^-$ ($T_{l:m}$) | 42.76 | 20.92 | 5.79 |
| | Grok v3 One-shot ($T_l$) + CoG ($T_{l:m}$) | 48.57 | 31.08 | 8.07 |
| | Grok v3 One-shot$^-$ ($T_l$) + CoG$^-$ ($T_{l:m}$) | 41.75 | 19.08 | 5.21 |
| **Three pieces** | GPT-4o One-shot ($T_l$) + CoG ($T_{l:m}$) + CoG | 48.46 | 31.07 | 8.42 |
| | GPT-4o One-shot$^-$ ($T_l$) + CoG$^-$ ($T_{l:m}$) + CoG ($T$) | 49.53 | 33.54 | 9.38 |
| | Gemini 2.5 Pro One-shot ($T_l$) + CoG ($T_{l:m}$) + CoG ($T$) | 50.64 | 35.38 | **10.16** |
| | Gemini 2.5 Pro One-shot$^-$ ($T_l$) + CoG$^-$ ($T_{l:m}$) + CoG ($T$) | 49.63 | 34.15 | 9.28 |
| | Grok v3 One-shot ($T_l$) + CoG ($T_{l:m}$) + CoG ($T$) | **50.91** | **36.62** | 10 |
| | Grok v3 One-shot$^-$ ($T_l$) + CoG$^-$ ($T_{l:m}$) + CoG ($T$) | 50.41 | 35.69 | 9.70 |

# B    SYSTEM PROMPTS DESIGN

Prompt segmentation plays a critical role in our CoG framework. The effectiveness of chain-of-generation heavily depends on whether LLMs can correctly understand a molecule's natural language description and decompose it into components following a hierarchical order. When LLMs successfully perform this segmentation, identifying and separating structural elements in the correct sequence leads to better overall performance. As such, the design of the system prompt provided to the LLM is crucial.

In this section, we compare two designs of system prompts: one with detailed component definitions and one without them ($^-$). As shown in Table 3, Grok v3 with the detailed component definitions shows a stable performance. The performance gap across different LLMs can be attributed to their varying abilities to understand the text description and accurately segment it into a coarse-to-fine hierarchy.

```
System Prompt without detailed instruction:

You are a highly capable AI tasked with generating four text outputs (A
only, AB, C only, BC) by segmenting a single molecule's text description
provided as {DESCRIPTION}. The description includes three components (A,
B, C) in a syntactic form such as "The molecule contains..." or "The
molecule is made of...". Your task is to analyze the description,
identify components A, B, and C based on their roles, and generate four
outputs with modified text descriptions containing the specified
component subsets.

Output Generation Rules:

A only: Include only the A component in the text description.
AB: Include both A and B components in the text description.
C only: Include only the C component in the text description.
BC: Include both B and C components in the text description.

Instructions:

Read the text description {DESCRIPTION} to identify A, B, and C based on
their significance in the molecule.
Write the new description in a consistent format, e.g., "The molecule is
made of [component(s)]", ensuring grammatical correctness.
If components are ambiguous, prioritize the most significant part as A,
the next as B, and the least significant as C.
```

```
System Prompt with detailed instruction:

You are a highly capable AI tasked with generating four text outputs (A
only, AB, C only, BC) by segmenting a single molecule's text description
provided as {DESCRIPTION}. The description includes three components (A,
B, C) in a syntactic form such as "The molecule contains..." or "The
molecule is made of...". Your task is to analyze the description, extract
 the components A, B, and C based on their semantic roles, and generate
four outputs with modified text descriptions containing the specified
component subsets. The output should include only the modified text
descriptions.

Component Definitions:

A: The main structural core or primary ring system (e.g., benzene ring,
naphthalene ring, pyrimidine ring) or the most central atom in smaller
molecules (e.g., carbon, nitrogen, phosphorus).
B: Medium-sized functional groups or secondary ring systems directly
attached to the core (e.g., amide group, phosphate group, nitro group,
cyclohexane ring).
C: Small atomic or functional group modifiers attached to A or B (e.g.,
fluorine, chlorine, bromine, iodine, hydroxyl group).

Output Generation Rules:

From heavy to light:
A only: Include only the A component in the text description.
AB: Include both A and B components in the text description.
From light to heavy:
C only: Include only the C component in the text description.
BC: Include both B and C components in the text description.
```

```
702
703   For the input description:
704   Read the text description {DESCRIPTION} to identify A, B, and C based on
705   their structural significance, regardless of the syntactic structure.
706   Write the new description in a consistent format, e.g., "The molecule is
      made of [component(s)]", ensuring grammatical correctness.
707   If a component appears ambiguous, prioritize the largest ring system or
708   core as A, the next significant group as B, and the smallest modifier as
709   C.
710
711
712   System Prompt on real datasets:
713
714   You are a highly capable AI tasked with analyzing a molecule's text
      description provided as {DESCRIPTION} and outputting a single number
715   representing the count of distinct components, along with increment of
716   textual segmentations. The description may include phrases like "The
717   molecule is made of..." or similar syntactic forms describing molecular
718   components. Your task is to segment the description into meaningful
719   components based on their semantic roles and provide the reasoning for
      your segmentation.
720
721   Component Identification Guidelines
722
723   To determine the number of components, consider the following candidate
724   definitions as a flexible framework. Use your reasoning to adapt these
      definitions or create new ones based on the specific description provided:
725
726
727   Core Structure: The primary scaffold or backbone (e.g., glutamyl-alanine,
       tyrosine, glycerone) counts as 1 component.
728   Functional Groups: Each distinct functional group (e.g., hydroxy, methyl,
729    oxo, amino) is counted separately, even if multiple instances exist (e.g
730   ., three hydroxy groups = three components).
731   Modifiers: Each distinct modifier (e.g., ionic forms, complex
732   substituents like methylthioethyl, stereochemical configurations) is
      counted separately.
733   Stereochemistry: Enantiomeric or stereochemical configurations (e.g., L-
734   enantiomer, (2S,4S)-configuration) are counted as modifiers if explicitly
735    described as part of the molecule's structure.
736   Chemical Properties: Descriptive attributes of the molecule, such as
737   acidity, basicity, or physical state (e.g., solid, liquid, gas).
738   You are free to refine or reinterpret these categories based on the
      description's context.
739
740   Output Format:
741   For a molecule with n components, the output line includes:
742   The integer n (total component count).
743   A tab.
744   A fluent sentence for part 1 (core).
745   A tab.
      A fluent sentence for parts 1+2 (core + first modifier/functional group).
746   A tab.
747   A fluent sentence for parts 1+2+3, and so on, up to all n parts.
748   Sentences are concise, chemically accurate, and cumulative, incorporating
       all prior components in each subsequent sentence.
749
750   Component Counting: Each core, functional group, and modifier was counted
       separately, without grouping similar substituents.
751
752
753   Example:
754   Input: "The molecule is a 1,8-naphthyridine derivative that is 4-oxo-1,4-
      dihydro-1,8-naphthyridine-3-carboxylic acid bearing additional 2,4-
755   difluorophenyl, fluoro and 3-aminopyrrolidin-1-yl substituents at
      positions 1, 6 and 7 respectively. It is a 1,8-naphthyridine derivative,
```

```
an amino acid, a monocarboxylic acid, an organofluorine compound, an
aminopyrrolidine, a tertiary amino compound, a primary amino compound and
 a quinolone antibiotic. It is a conjugate base of a 1-[6-carboxy-8-(2,4-
difluorophenyl)-3-fluoro-5-oxo-5,8-dihydro-1,8-naphthyridin-2-yl]
pyrrolidin-3-aminium."

Output:
Number: 5
Segmentations: The molecule is a heterocycle with a naphthyridine core.
The molecule is a heterocycle with a naphthyridine core, substituted by a
 2,4-difluorophenyl group at position 1.
The molecule is a heterocycle with a naphthyridine core, substituted by a
 2,4-difluorophenyl group at position 1 and a fluoro group at position 6.

The molecule is a heterocycle with a naphthyridine core, substituted by a
 2,4-difluorophenyl group at position 1, a fluoro group at position 6,
and a 3-aminopyrrolidin-1-yl group at position 7.
The molecule is a heterocycle with a naphthyridine core, substituted by a
 2,4-difluorophenyl group at position 1, a fluoro group at position 6, a
3-aminopyrrolidin-1-yl group at position 7, and a carboxy group at
position 3 to form a quinolone antibiotic.

Constraints:
Ensure the segmentation is concise, chemically informed, fluent, and
directly tied to the description. If the description is ambiguous, make
reasonable assumptions.
```

## C  EVALUATION METRICS

We derive four final metrics from basic proportions of generated molecules $G'$ that satisfy different levels of MACCS Tanimoto similarity with the ground truth graph $G$, as well as their pairwise distinctness. Specifically, we define the following intermediate proportions:

(1) $p_{\text{base}}$: proportion of generated molecules with $f(G', G) > 0.5$, where $f$ denotes the MACCS Tanimoto similarity, representing molecules that satisfy the basic textual fidelity requirements. (2) $p_{\text{qual}}$: preferred proportion of generated molecules if if $f(G', G)$ in $(0.5, 0.8)$, which indicates that the qualified molecules meet textual requirements while still retaining freedom and creativity. (3) $p_{\text{dist}}$: evaluates the structural variability among generated outputs. It is computed as the average pairwise distance $1 - f(G'_i, G'_j)$ between all molecules that broadly satisfy the textual fidelity. (4) $p_{\text{val}}$: proportion of chemically valid molecules among all generated outputs, determined by standard chemistry rules.

Based on these quantities, we define three composite evaluation metrics. Qualified Novelty(Q-Nov) is computed as $p_{\text{qual}} \times p_{\text{dist}}$, capturing the extent to which the generated molecules simultaneously exhibit high fidelity, novelty, and mutual distinctness. Qualified Coverage(Q-Cov) is defined as $p_{\text{val}} \times p_{\text{qual}}$, reflecting how well valid molecules extend into new chemical space. Finally, the Balanced Quality Index (BQI) integrates all dimensions into a single score through a equally weighted combination of $p_{\text{val}}, p_{\text{base}}, p_{\text{dist}},$ and $p_{\text{qual}}$.

## D  PERFORMANCE ON BENCHMARK SUBSETS

To apply our CoG strategy on benchmark datasets, we first define segmentation rules with three stages: core structure, functional groups and modifiers, and stereochemical configurations. To fully capture the chemical information contained in the often complex and unnormalized textual prompts, we leverage Grok v3 and its reasoning capabilities to parse and segment each prompt, as illustrated in Appendix B. As shown in Fig. 5 and Fig. 6, most prompts can be segmented into two or three pieces, successfully extracting the core structure, functional groups and modifiers, and stereochemical configurations. Results across different segmentation subsets are summarized in Tables 4 to 10.

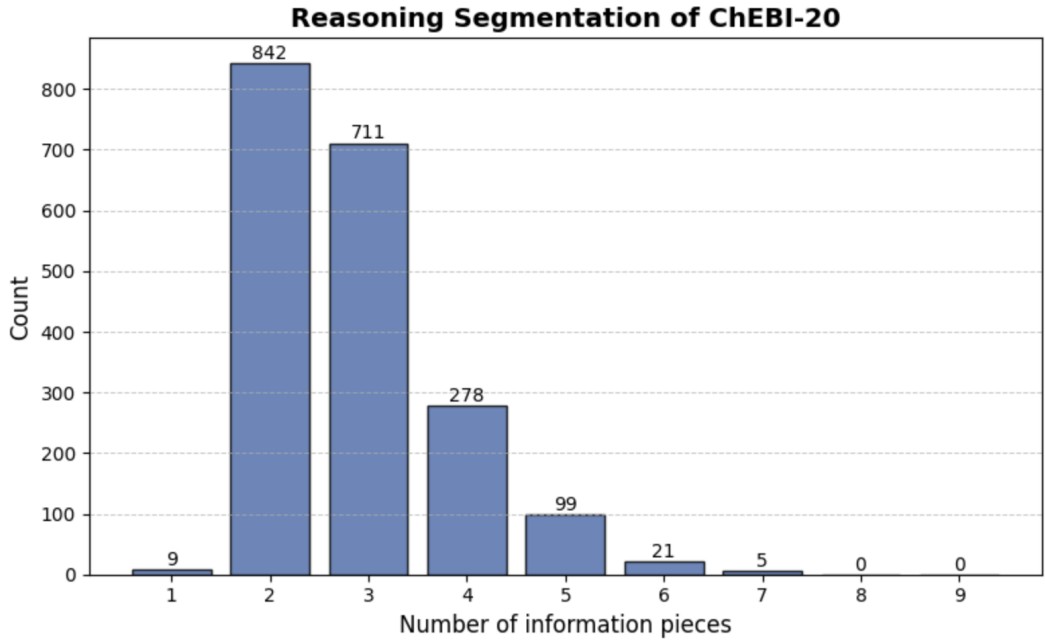

Figure 5: Distribution of number of information pieces after reasoning segmentation on ChEBI-20

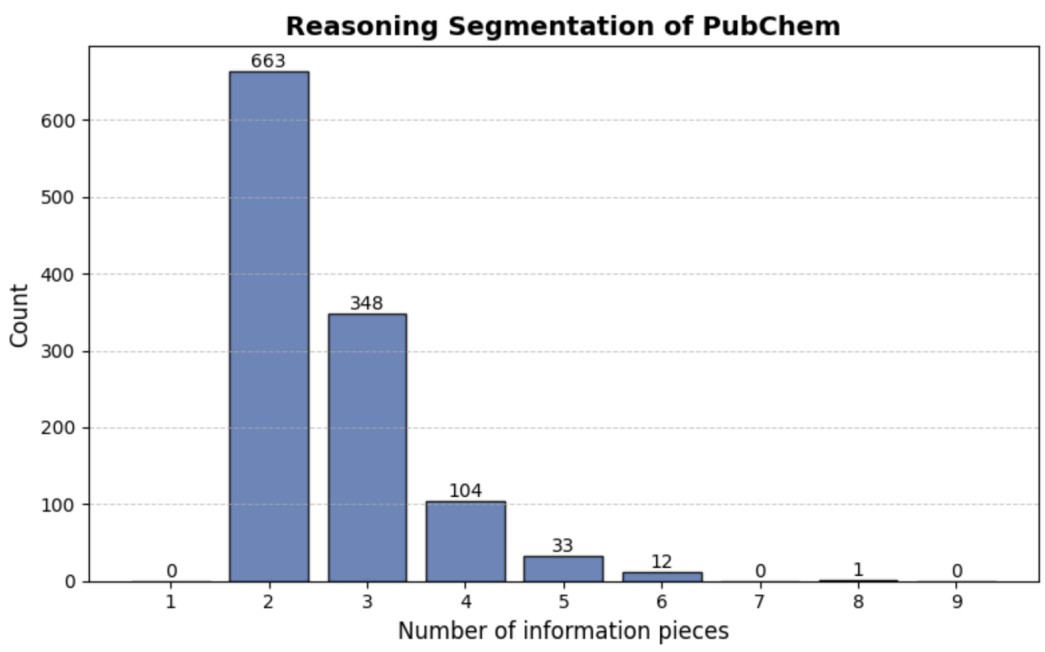

Figure 6: Distribution of number of information pieces after reasoning segmentation on PubChem

## E    VISUALIZATION OF GENERATED SAMPLES

Table 4: Performance on the ChEBI-20 subset where each textual prompt contains five pieces of information. All values are reported as percentages.

| Method | BQI | Q-Cov | Q-Nov | Validity |
|---|---|---|---|---|
| **SMILES-based methods** | | | | |
| MolT5 small | 51.37 | 24.49 | 7.07 | 66.33 |
| MolT5 large | 57.56 | 20.41 | 3.04 | 97.96 |
| ChemT5 small | 61.05 | 30.61 | 5.85 | 94.9 |
| ChemT5 base | 58.79 | 22.45 | 4.46 | 95.92 |
| BioT5 base | 67.24 | 34.29 | 13.22 | **100** |
| BioT5 plus | 63.38 | 26.73 | 7.32 | **100** |
| **Graph-based methods** | | | | |
| 3M diffusion | 59.9 | 44.29 | 12.84 | **100** |
| GraphLDM | 67.77 | 58.16 | **17.49** | **100** |
| GraphLDM + CoG | **68.42** | **59.39** | 16.98 | **100** |

Table 5: Performance on the ChEBI-20 subset where each textual prompt contains four pieces of information. All values are reported as percentages.

| Method | BQI | Q-Cov | Q-Nov | Validity |
|---|---|---|---|---|
| **SMILES-based methods** | | | | |
| MolT5 small | 53.25 | 27.79 | 7.84 | 68.95 |
| MolT5 large | 59.8 | 29.24 | 4.39 | 97.11 |
| ChemT5 small | 59.63 | 27.8 | 4.79 | 96.03 |
| ChemT5 base | 60.55 | 28.88 | 5.19 | 96.39 |
| BioT5 base | **64.77** | 31.19 | 9.62 | **100** |
| BioT5 plus | 59.89 | 19.71 | 4.31 | **100** |
| **Graph-based methods** | | | | |
| 3M diffusion | 58.14 | 43.68 | 11.63 | **100** |
| GraphLDM | 63.19 | 50.4 | **14.8** | **100** |
| GraphLDM + CoG | 63.7 | **51.55** | 13.81 | **100** |

Table 6: Performance on the ChEBI-20 subset where each textual prompt contains three pieces of information. All values are reported as percentages.

| Method | BQI | Q-Cov | Q-Nov | Validity |
|---|---|---|---|---|
| **SMILES-based methods** | | | | |
| MolT5 small | 52.61 | 26.06 | 5.47 | 77.61 |
| MolT5 large | 56.08 | 20.28 | 2.29 | 98.03 |
| ChemT5 small | 57.9 | 23.66 | 3.26 | 97.04 |
| ChemT5 base | 58.28 | 25.92 | 3.6 | 97.18 |
| BioT5 base | 59.31 | 20.9 | 4.2 | **100** |
| BioT5 plus | 57.23 | 15.71 | 2.47 | **100** |
| **Graph-based methods** | | | | |
| 3M diffusion | 56.34 | 40.23 | 10.06 | **100** |
| GraphLDM | 59.75 | 43.44 | **14.8** | **100** |
| GraphLDM + CoG | **60.7** | **45.38** | 11.44 | **100** |

Table 7: Performance on the ChEBI-20 subset where each textual prompt contains two pieces of information. All values are reported as percentages.

| Method | BQI | Q-Cov | Q-Nov | Validity |
|---|---|---|---|---|
| **SMILES-based methods** | | | | |
| MolT5 small | 49.88 | 21.52 | 3.33 | 83.71 |
| MolT5 large | 54.39 | 17.47 | 1.38 | 98.45 |
| ChemT5 small | 56.28 | 22 | 2.43 | 97.5 |
| ChemT5 base | 56.53 | 23.43 | 2.64 | 97.74 |
| BioT5 base | 56.12 | 15.91 | 2.21 | 99.97 |
| BioT5 plus | 54.56 | 12.49 | 1.3 | **100** |
| **Graph-based methods** | | | | |
| 3M diffusion | 55.25 | 38.83 | 8.81 | **100** |
| GraphLDM | 56.94 | 38.53 | 9.1 | **100** |
| GraphLDM + CoG | **58.32** | **42.28** | **9.23** | **100** |

Table 8: Performance on the PubChem subset where each textual prompt contains four pieces of information. All values are reported as percentages.

| Method | BQI | Q-Cov | Q-Nov | Validity |
|---|---|---|---|---|
| **SMILES-based methods** | | | | |
| MolT5 small | 48.59 | 14.56 | 1.01 | 81.55 |
| MolT5 large | 53.52 | 13.6 | 0.85 | 99.03 |
| ChemT5 small | 54.78 | 17.48 | 1.95 | 94.17 |
| ChemT5 base | 56.88 | 20.39 | 2.82 | 97.09 |
| BioT5 base | 56.38 | 12.82 | 1.95 | **100** |
| BioT5 plus | 54.63 | 9.51 | 1.24 | **100** |
| **Graph-based methods** | | | | |
| 3M diffusion | 59.98 | **49.51** | 13.04 | **100** |
| GraphLDM | 58.79 | 41.17 | 12.96 | **100** |
| GraphLDM + CoG | **60.88** | 48.16 | **14.69** | **100** |

Table 9: Performance on the PubChem subset where each textual prompt contains three pieces of information. All values are reported as percentages.

| Method | BQI | Q-Cov | Q-Nov | Validity |
|---|---|---|---|---|
| **SMILES-based methods** | | | | |
| MolT5 small | 47.98 | 12.1 | 0.98 | 80.4 |
| MolT5 large | 52.31 | 11.23 | 0.65 | 97.69 |
| ChemT5 small | 54.87 | 17.58 | 1.79 | 97.12 |
| ChemT5 base | 55.8 | 20.46 | 2.21 | 95.39 |
| BioT5 base | 55.23 | 12.85 | 1.71 | **100** |
| BioT5 plus | 54.92 | 11.56 | 1.39 | **100** |
| **Graph-based methods** | | | | |
| 3M diffusion | 56.48 | 41.67 | 11.31 | **100** |
| GraphLDM | **59.06** | 43.4 | **13.66** | **100** |
| GraphLDM + CoG | 58.92 | **44.84** | 12.25 | **100** |

Table 10: Performance on the PubChem subset where each textual prompt contains two pieces of information. All values are reported as percentages.

| Method | BQI | Q-Cov | Q-Nov | Validity |
|---|---|---|---|---|
| **SMILES-based methods** | | | | |
| MolT5 small | 41.58 | 9.67 | 1.08 | 67.67 |
| MolT5 large | 50.66 | 19.33 | 1.96 | 91.54 |
| ChemT5 small | 51.22 | 21.75 | 2.9 | 88.82 |
| ChemT5 base | 50.88 | 21 | 3.03 | 87.16 |
| BioT5 base | 54.22 | 19.46 | 3.66 | **100** |
| BioT5 plus | 55.19 | 21.21 | 4.25 | **100** |
| | | | | |
| **Graph-based methods** | | | | |
| 3M diffusion | 53.94 | 37.76 | 10.15 | **100** |
| GraphLDM | 56.97 | 39.52 | **11.92** | **100** |
| GraphLDM + CoG | **57.36** | **41.84** | 11.37 | **100** |

Figure 7: Sequence-to-sequence methods struggle to produce valid and diverse outputs when given prompts such as "The molecule is a simple structure that contains a benzene ring." In particular, ChemT5 and MolT5 exhibit severe mode collapse: despite the large solution space implied by flexible prompts, they repeatedly converge to a single rigid output. A second limitation lies in chemical invalidity, as exemplified by the ChemT5-base augment model.

