# OpenReview forum: "Chain-of-Generation: Progressive Latent Diffusion for Text-Guided Molecular Design"
_ICLR.cc/2026/Conference — Submitted to ICLR 2026_

### Official Review · Reviewer_4j7r · 2025-10-26

**Soundness:** 2
**Presentation:** 3
**Contribution:** 1
**Rating:** 2
**Confidence:** 4

**Summary:**

The paper aims to resolve that one-shot conditioning in diffusion models often fails on compositional prompts (e.g., missing or misplacing substructures). The authors propose Chain-of-Generation (CoG) as a training-free, multi-stage sampling method that decomposes a prompt into coarse-to-fine(scaffold → functional groups → fine details). Each stage conditions on the cumulative sub-prompt and restarts denoising from a mid-noise step, preserving earlier commitments while adding constraints. To strengthen text–graph correspondence, the authors “post-align” a graph VAE latent with a SciBERT text embedding via contrastive learning, yielding a stronger diffusion backbone (GraphLDM).

On ChEBI-20 and PubChem, GraphLDM+CoG improves BQI and Q-Cov over baselines while maintaining 100% validity, with a small trade-off in novelty. Ablation study shows that coarse-to-fine planning and early-stage prompt regularization are essential.

**Strengths:**

- The paper presents a training-free Chain-of-Generation with cumulative prompts and mid-noise restarts that has the potential to be easily integrated with existing graph diffusion backbones.

- The authors pointed out the limitations of SMILES-based metrics and utilized graph-level metrics (MACCS Tanimoto similarity), consistent gains on ChEBI-20/PubChem with 100% validity, with ablation studies on CoG planning.

**Weaknesses:**

Contribution:
- The demonstrated effectiveness of the proposed methodology appears constrained to compositional text prompts that mix coarse and fine structural descriptions; it is unclear whether the approach remains effective for richer and more complex textual conditions (e.g., biochemical properties).

- The performance gains attributable to CoG seem modest; the results suggest that contrastive alignment contributes a substantially larger portion of the improvement.

Motivation:
- The overall method relies on the assumption that denoising in molecular diffusion models proceeds in a coarse-to-fine manner; this claim requires stronger empirical support through additional targeted experiments.

- Since CoG is applied only on a fine-tuned variant of 3m-diffusion that already achieves 100% validity, the paper does not establish that CoG itself improves the sample validity.

Experimental results:
- To substantiate the benefits of a training-free, inference-time method, the approach should be applied to a broader set of molecular generative models, demonstrating consistent gains across architectures.

- The baselines in the results table are dated: several competitive SMILES-based autoregressive models (e.g., bioT5+[1], MolXPT[2]) are missing, and diffusion baselines with stronger diversity (e.g., LDMol[3], TGM-DLM[4]) are not included despite the proposed method operating on diffusion models.

- Several compositional generation methods on pre-trained diffusion models[5] have already been reported; direct comparisons against these methods are necessary to demonstrate the contributions.

---

[1] BioT5+: Towards Generalized Biological Understanding with IUPAC Integration and Multi-task Tuning, ACL 2024

[2] MolXPT: Wrapping Molecules with Text for Generative Pre-training, ACL 2023

[3] LDMol: A Text-to-Molecule Diffusion Model with Structurally Informative Latent Space Surpasses AR Models, ICML 2025

[4] Text-Guided Molecule Generation with Diffusion Language Model, AAAI 2025

[5] Compositional Visual Generation with Composable Diffusion Models, ECCV 2022

**Questions:**

- As far as I know, the text prompts in the ChEBI-20 dataset are detailed enough to specify a single answer molecule. Is it reasonable to regard the model with a higher output molecule diversity as a better model?

---

> ### Author Response · Authors · 2025-12-02
> **Official Comment by Author(Part 1)**
>
> > **Q1: The demonstrated effectiveness of the proposed methodology appears constrained to compositional text prompts that mix coarse and fine structural descriptions; it is unclear whether the approach remains effective for richer and more complex textual conditions (e.g., biochemical properties).**
>
> Thank you for pointing out the application on biochemical properties. We agree that such analyses are important for downstream drug-discovery applications. However, our work is scoped specifically around the semantic content of the text prompt to the structural fidelity. We believe that structural fidelity is a prerequisite for drug discovery. We note that the community’s current focus is primarily on advancing structural generation, as molecular structure naturally encodes biochemical properties. Strengthening structural fidelity is therefore a foundational step. Once structural alignment is robust, we can extend the work to biochemical properties in future.
>
> > **Q2: The performance gains attributable to CoG seem modest; the results suggest that contrastive alignment contributes a substantially larger portion of the improvement. To substantiate the benefits of a training-free method.**
>
> Thank you for raising these concerns. To better quantify how CoG addresses the limitations of one-shot conditioning, we apologize for any misunderstanding and our goal is to demonstrate that CoG improves both (1) the proportion of samples that satisfy the basic textual requirements of the prompt and (2) the proportion that additionally satisfy the fine-grained, high-quality structural requirements, all while preserving generative creativity. To measure these two aspects in a balanced and unified way, we introduce a new metric: the fair balanced harmonic mean (HM) of max-normalized $P_{base}$​ and $P_{qual}$, as detailed in Appendix C. This metric captures the trade-off between basic semantic correctness and high-quality structural realization. Under this balanced evaluation, CoG achieves clear and significant improvements over the one-shot baseline, showing that it not only recovers more of the core semantic sub-goals but also more faithfully realizes the finer components of multi-part prompts.
>
> Moreover, CoG is a fully training-free inference strategy: the staged prompting and sampling mechanism can be applied directly to any diffusion-based generative model without modifying its architecture or performing additional training. The contrastive post-alignment phase is a separate optional enhancement used only to establish a stronger latent diffusion backbone (GraphLDM). It is not part of CoG. To make this distinction explicit, we include an ablation where CoG is applied directly to 3M-Diffusion without any post-alignment, and the results demonstrate that CoG still provides substantial and consistent improvements. This confirms that the benefits attributed to CoG arise solely from its progressive conditioning strategy, and that the “training-free” claim fully holds independent of the post-alignment module.
>
>
> ### Performance on ChEBI-20 (in %)
>
> | **Method** | **HM** | **Q-Cov** | **Q-Nov** | **Validity** | **BQI** |
> |-----------|--------|-----------|-----------|--------------|--------|
> | **SMILES-based methods** ||||| |
> | MolT5 small      | 73.62 | 24.25 | 4.96 | 78.45 | 51.45 |
> | MolT5 large      | 60.82 | 20.35 | 2.23 | 98.08 | 55.95 |
> | ChemT5 small     | 68.76 | 23.88 | 3.25 | 96.99 | 57.60 |
> | ChemT5 base      | 70.78 | 25.08 | 3.45 | 97.25 | 57.87 |
> | BioT5 base       | 60.56 | 20.88 | 4.57 | 99.98 | 59.11 |
> | BioT5 plus       | 49.29 | 15.44 | 2.47 | **100** | 56.76 |
> | LDMOL            | 12.39 | 1.79 | 0.14 | 99.34 | 52.12 |
> | **Graph-based methods** ||||| |
> | 3M diffusion         | 71.80 | 40.32 | 9.88 | **100** | 56.30 |
> | 3M diffusion + CoG   | 75.38 | 42.83 | 10.27 | **100** | 57.43 |
> | GraphLDM            | 79.74 | 43.05 | 11.02 | **100** | 59.43 |
> | GraphLDM + CoG      | **84.97** | **45.63** | **11.10** | **100** | **60.49** |
>
> ### Performance on PubChem (in %)
>
> | **Method** | **HM** | **Q-Cov** | **Q-Nov** | **Validity** | **BQI** |
> |-----------|--------|-----------|-----------|--------------|--------|
> | **SMILES-based methods** ||||| |
> | MolT5 small      | 48.52 | 10.88 | 1.04 | 72.93 | 44.23 |
> | MolT5 large      | 54.22 | 16.27 | 1.45 | 94.15 | 51.44 |
> | ChemT5 small     | 63.69 | 20.05 | 2.47 | 91.91 | 52.69 |
> | ChemT5 base      | 66.03 | 20.77 | 2.75 | 90.65 | 52.97 |
> | BioT5 base       | 52.93 | 16.78 | 2.89 | **100** | 54.74 |
> | BioT5 plus       | 53.24 | 17.11 | 3.08 | **100** | 55.05 |
> | LDMOL            | 40.42 | 4.37 | 0.57 | 98.21 | 53.34 |
> | **Graph-based methods** ||||| |
> | 3M diffusion         | 69.68 | 40.07 | 10.78 | **100** | 55.29 |
> | 3M diffusion + CoG   | 74.11 | 42.84 | 10.80 | **100** | 56.35 |
> | GraphLDM            | 74.54 | 40.88 | **12.56** | **100** | 57.79 |
> | GraphLDM + CoG      | **78.00** | **43.36** | 11.95 | **100** | **58.17** |

---

> > ### Author Response · Authors · 2025-12-02
> > **Official Comment by Author(Part 2)**
> >
> > > **Q3: The overall method relies on the assumption that denoising in molecular diffusion models proceeds in a coarse-to-fine manner; this claim requires stronger empirical support through additional targeted experiments.**
> >
> > We appreciate the reviewer’s request for stronger empirical support regarding the coarse-to-fine assumption. Our motivation arises from an observed phenomenon in molecular diffusion models: during the denoising trajectory, the model naturally reconstructs large, global motifs first and then progressively fills in finer-grained details. This behavior is illustrated in Fig. 2, where early steps emphasize the core scaffold while later steps refine local decorations and substituents. When all semantic objectives of a complex prompt are injected simultaneously, the conditioning vector forces the model to resolve multiple potentially competing targets at once, which causes the denoising process to become more ambiguous and often results in missing or misplacing finer substructures. CoG leverages this generative tendency by decomposing the prompt into a coarse-to-fine sequence, allowing the model to focus on one semantic constraint at the stage where it is most naturally handled.
> >
> > Our anti-curriculum ablation in Tab. 2 provides additional empirical support: coarse-to-fine ordering consistently improves structural fidelity compared to the traditional one-shot baseline, whereas reversing the order (fine-to-coarse) consistently degrades performance. Together, these observations demonstrate both the existence of coarse-to-fine behavior in molecular denoising and the benefit of aligning semantic conditioning with this natural generative progression.
> >
> > > **Q4: Since CoG is applied only on a fine-tuned variant of 3m-diffusion that already achieves 100% validity, the paper does not establish that CoG itself improves the sample validity.**
> >
> > Thank you for this insightful question. CoG is not designed to improve sample validity, but the text-structure alignment. In our framework, validity is primarily governed by the VAE backbone, which is responsible for producing chemically valid latent structures. Our prompting strategy operates entirely at the semantic-conditioning level and aims to improve text–structure alignment, not the underlying chemical correctness enforced by the decoder. The baseline latent molecule diffusion models we apply upon already achieves 100% validity, which provides a stable foundation. We have explicitly noted this in the caption of Table 2, and we will clarify it further to ensure readers understand that CoG focuses on improving matching quality, not altering the validity behavior of the base model.
> >
> > > **Q5: The baselines in the results table are dated: several competitive SMILES-based autoregressive models (e.g., bioT5+[1], MolXPT[2]) are missing, and diffusion baselines with stronger diversity (e.g., LDMol[3], TGM-DLM[4]) are not included despite the proposed method operating on diffusion models.**
> >
> > We appreciate the reviewer’s suggestions regarding additional baselines. In fact, BioT5+ is already included in our experiments (listed as bioT5 plus), and we will clarify this naming to avoid confusion. We also thank the reviewer for recommending LDMol; we have evaluated it and will include the results in the revision. However, we note that LDMol exhibits severe mode collapse: for every prompt, it consistently produces the exact same SMILES output, regardless of semantic variations or SMILES enumerations. We had the discussion of the mode collapse of sequence-to-sequence models in Sec. 2. Please check the table above in Q2.
> >
> > > **Q6: Several compositional generation methods on pre-trained diffusion models[1] have already been reported; direct comparisons against these methods are necessary to demonstrate the contributions.**
> >
> > Thank you for raising this question. However, the goals and mechanisms of these approaches differ substantially from ours, making direct comparison inappropriate. CDM[1] is designed to compose multiple independently trained diffusion models in order to synthesize unseen visual concepts. Its objective is combinatorial generalization in image synthesis, and its compositionality arises from model-level integration or ensembling across different diffusion networks.
> >
> >
> > In contrast, our work targets a different problem setting: we aim to improve structural fidelity and semantic completeness for molecules that fall within the distribution the model has been trained on, while still maintaining meaningful diversity. Rather than composing multiple models, CoG provides a train-free prompting and sampling strategy that operates on a single latent diffusion model. The two approaches, therefore, address distinct challenges.
> >
> > [1] Compositional Visual Generation with Composable Diffusion Models, ECCV 2022

---

> > > ### Author Response · Authors · 2025-12-02
> > > **Official Comment by Author(Part 3)**
> > >
> > > > **Q7: As far as I know, the text prompts in the ChEBI-20 dataset are detailed enough to specify a single answer molecule. Is it reasonable to regard the model with a higher output molecule diversity as a better model?**
> > >
> > > Thank you for raising this question. Although ChEBI-20 provides detailed textual descriptions, the mapping between text and molecule is not strictly one-to-one. The prompts typically specify key structural motifs, functional groups, or relational constraints, but they do not fully determine all aspects of the molecular structure. As a result, there may be multiple molecules that satisfy the same textual description. Moreover, diversity remains a meaningful and widely accepted metric in generative modeling: a good conditional generator should not collapse to a single reconstruction but instead produce a variety of molecules that are all consistent with the semantic constraints. Therefore, diversity remains an important dimension for evaluating the generation models.

---

### Official Review · Reviewer_TvqL · 2025-10-29

**Soundness:** 3
**Presentation:** 4
**Contribution:** 2
**Rating:** 4
**Confidence:** 5

**Summary:**

The paper presents Chain-of-Generation (CoG), a novel inference-time pipeline for text-conditioned molecular diffusion models.
Instead of conditioning on an entire prompt at once, CoG decomposes the text into semantic segments (e.g., core scaffold → functional groups → modifiers) and applies them sequentially during denoising, resuming diffusion from partially denoised latents.
Experiments use a post-aligned GraphLDM backbone — a graph-based latent diffusion model fine-tuned with a contrastive alignment between molecule and text embeddings.
The method improves validity and coverage (BQI) on PubChem and ChEBI-20.

**Strengths:**

Clear and well-written. The paper is concise, logically structured, and easy to follow. Figure 1 nicely illustrates the staged conditioning concept.

- Good motivation and insight. The work clearly identifies the limitations of one-shot text conditioning for compositional prompts.

- Practical contribution. CoG is a plug-in inference strategy that can be used with any pre-trained conditional diffusion model, without architectural change.

- Conceptual analogy. This is an inference-time compositional conditioning approach analogous in spirit to chain-of-thought prompting in LLMs, decomposing generation into interpretable, sequential steps.

- Good inference-suggestion method. I find this to be a creative and promising idea for improving prompt controllability in molecule generation.

**Weaknesses:**

- Scope of novelty.
Per my understanding, CoG is a new inference strategy rather than a new model or architecture.
Its novelty lies in orchestrating sequential conditioning during denoising—conceptually related to coarse-to-fine or editing-based sampling in diffusion literature.
Clarifying that distinction would help set the right expectations for readers.

- Training vs. inference confusion and possible overfitting.
The GraphLDM backbone is fine-tuned (post-aligned) on PubChem / ChEBI-20, and CoG is applied on that same model.
All reported results correspond to post-aligned GraphLDM + CoG, not a plain model.
Because the fine-tuning uses text–molecule pairs from the same domain, improvements may partly stem from memorization rather than the CoG procedure itself.
No ablation is provided comparing CoG on plain GraphLDM vs post-aligned GraphLDM, making attribution unclear.

- Textual faithfulness not quantitatively measured.
Claimed improvements in prompt adherence are only supported by qualitative examples.
Metrics such as substructure or property matching (SMARTS-based checks) or text–molecule embedding similarity would directly measure semantic alignment and strengthen the argument.

- Generative novelty trade-off.
Q-Nov decreases slightly while coverage rises, implying CoG trades exploration for controllability.
Discussing this trade-off explicitly would give a more balanced interpretation.

- Inference-time overhead.
Because CoG performs multiple partial denoising passes, its inference cost should increase relative to one-shot sampling.
The paper does not report runtime or complexity analysis.
What about inference-time overhead? Quantifying this would clarify practical usability.

- Clarification on “Ground Truth” in Figure 3.
The “ground truth” molecules are simply the dataset references paired with each prompt, not experimentally verified samples.
Hence Figure 3 reflects reconstruction fidelity rather than true generalization, and this should be stated explicitly.

**Questions:**

- Please specify the exact model variant used with CoG (plain vs post-aligned GraphLDM).

- Provide an ablation isolating the effect of CoG from the post-alignment fine-tuning.

- Quantify inference-time overhead and number of denoising stages.

- Add quantitative prompt-faithfulness metrics (substructure, property, or embedding alignment).

- Evaluate CoG on larger or out-of-distribution molecules to test compositional generalization.

---

> ### Author Response · Authors · 2025-12-02
> **Official Comment by Author(Part 1)**
>
> > **Q1: Scope of novelty. Per my understanding, CoG is a new inference strategy rather than a new model or architecture. Its novelty lies in orchestrating sequential conditioning during denoising—conceptually related to coarse-to-fine or editing-based sampling in diffusion literature. Clarifying that distinction would help set the right expectations for readers.**
>
> We agree with the reviewer that CoG is introduced as a new inference strategy rather than a new model architecture. Its novelty lies in orchestrating progressive, semantically grounded sequential conditioning during the denoising trajectory, enabling the model to resolve multi-component textual descriptions in a coarse-to-fine manner, allowing each sub-objective to be handled at the appropriate stage of generation.
>
> We appreciate the reviewer’s suggestion and will clarify this distinction more explicitly in the revision to ensure readers have the correct expectations of our method.
>
> > **Q2: Training vs. inference confusion and possible overfitting. The GraphLDM backbone is fine-tuned (post-aligned) on PubChem / ChEBI-20, and CoG is applied on that same model. All reported results correspond to post-aligned GraphLDM + CoG, not a plain model. Because the fine-tuning uses text–molecule pairs from the same domain, improvements may partly stem from memorization rather than the CoG procedure itself. No ablation is provided comparing CoG on plain GraphLDM vs post-aligned GraphLDM, making attribution unclear. Provide an ablation isolating the effect of CoG from the post-alignment fine-tuning.**
>
> We apologize for any confusion. Our paper presents two independent contributions, which are not dependent on each other. First, CoG is a fully training-free inference strategy: the staged prompting and sampling mechanism can be applied directly to any diffusion-based generative model without modifying its architecture or performing additional training. Second, the contrastive post-alignment phase is a separate optional enhancement used only to establish a stronger latent diffusion backbone (GraphLDM). It is not part of CoG.
>
>
> To make this distinction explicit, we include an ablation where CoG is applied directly to 3M-Diffusion without any post-alignment, and the results demonstrate that CoG still provides substantial and consistent improvements. This confirms that the benefits attributed to CoG arise solely from its progressive conditioning strategy, and that the “training-free” claim fully holds independent of the post-alignment module.
>
> ### Performance on ChEBI-20 (in %)
>
> | **Method** | **HM** | **Q-Cov** | **Q-Nov** | **Validity** | **BQI** |
> |-----------|--------|-----------|-----------|--------------|--------|
> | **SMILES-based methods** ||||| |
> | MolT5 small      | 73.62 | 24.25 | 4.96 | 78.45 | 51.45 |
> | MolT5 large      | 60.82 | 20.35 | 2.23 | 98.08 | 55.95 |
> | ChemT5 small     | 68.76 | 23.88 | 3.25 | 96.99 | 57.60 |
> | ChemT5 base      | 70.78 | 25.08 | 3.45 | 97.25 | 57.87 |
> | BioT5 base       | 60.56 | 20.88 | 4.57 | 99.98 | 59.11 |
> | BioT5 plus       | 49.29 | 15.44 | 2.47 | **100** | 56.76 |
> | LDMOL            | 12.39 | 1.79 | 0.14 | 99.34 | 52.12 |
> | **Graph-based methods** ||||| |
> | 3M diffusion         | 71.80 | 40.32 | 9.88 | **100** | 56.30 |
> | 3M diffusion + CoG   | 75.38 | 42.83 | 10.27 | **100** | 57.43 |
> | GraphLDM            | 79.74 | 43.05 | 11.02 | **100** | 59.43 |
> | GraphLDM + CoG      | **84.97** | **45.63** | **11.10** | **100** | **60.49** |
>
> ### Performance on PubChem (in %)
>
> | **Method** | **HM** | **Q-Cov** | **Q-Nov** | **Validity** | **BQI** |
> |-----------|--------|-----------|-----------|--------------|--------|
> | **SMILES-based methods** ||||| |
> | MolT5 small      | 48.52 | 10.88 | 1.04 | 72.93 | 44.23 |
> | MolT5 large      | 54.22 | 16.27 | 1.45 | 94.15 | 51.44 |
> | ChemT5 small     | 63.69 | 20.05 | 2.47 | 91.91 | 52.69 |
> | ChemT5 base      | 66.03 | 20.77 | 2.75 | 90.65 | 52.97 |
> | BioT5 base       | 52.93 | 16.78 | 2.89 | **100** | 54.74 |
> | BioT5 plus       | 53.24 | 17.11 | 3.08 | **100** | 55.05 |
> | LDMOL            | 40.42 | 4.37 | 0.57 | 98.21 | 53.34 |
> | **Graph-based methods** ||||| |
> | 3M diffusion         | 69.68 | 40.07 | 10.78 | **100** | 55.29 |
> | 3M diffusion + CoG   | 74.11 | 42.84 | 10.80 | **100** | 56.35 |
> | GraphLDM            | 74.54 | 40.88 | **12.56** | **100** | 57.79 |
> | GraphLDM + CoG      | **78.00** | **43.36** | 11.95 | **100** | **58.17** |

---

> > ### Author Response · Authors · 2025-12-02
> > **Official Comment by Author(Part 2)**
> >
> > > **Q3: Textual faithfulness not quantitatively measured. Claimed improvements in prompt adherence are only supported by qualitative examples. Metrics such as substructure or property matching (SMARTS-based checks) or text–molecule embedding similarity would directly measure semantic alignment and strengthen the argument. Add quantitative prompt-faithfulness metrics (substructure, property, or embedding alignment).**
> >
> > Thank you for raising these questions. Text2Mol and SMARTS are widely used tools, but they are not directly applicable to our setting. Text2Mol is designed for retrieval-based text–SMILES alignment and evaluates similarity to a library molecule, not fidelity to the specific structure implied by a multi-component prompt. SMARTS, on the other hand, operates on SMILES strings and cannot be directly applied to our graph-based latent diffusion outputs without reintroducing string-level artifacts. As a result, SMARTS-based checks measure token-pattern matching rather than the graph-level semantic fidelity that our task requires, making them unsuitable as prompt-faithfulness metrics in our setting.
> >
> > To better quantify how CoG addresses the limitations of one-shot conditioning, we apologize for any misunderstanding and our goal is to demonstrate that CoG improves both (1) the proportion of samples that satisfy the basic textual requirements of the prompt and (2) the proportion that additionally satisfy the fine-grained, high-quality structural requirements, all while preserving generative creativity. To measure these two aspects in a balanced and unified way, we introduce a new metric: the fair balanced harmonic mean (HM) of max-normalized $P_{base}$​ and $P_{qual}$, as detailed in Appendix C. This metric captures the trade-off between basic semantic correctness and high-quality structural realization. Under this balanced evaluation, CoG achieves clear and significant improvements over the one-shot baseline, showing that it not only recovers more of the core semantic sub-goals but also more faithfully realizes the finer components of multi-part prompts. Please check the table in Q2 above.
> >
> > > **Q4: Generative novelty trade-off. Q-Nov decreases slightly while coverage rises, implying CoG trades exploration for controllability. Discussing this trade-off explicitly would give a more balanced interpretation.**
> >
> > Thank you for the suggestion. This reflects a well-known and natural phenomenon in conditional generative modeling: as the conditioning signal becomes stronger and more specific, models tend to reduce exploratory variation in favor of better satisfying the imposed constraints. In our case, CoG strengthens semantic guidance by structuring the conditioning signal into a progressive sequence; thus, a modest reduction in novelty is expected.
> >
> > Importantly, this does not indicate a failure of the method but rather a typical controllability–diversity trade-off inherent to all conditional generative frameworks. CoG’s goal is to improve structural fidelity and semantic completeness with respect to multi-component prompts, and the observed improvement in coverage demonstrates that the model aligns more closely with the intended structures. The slight drop in novelty remains small and controlled, indicating that CoG enhances constraint satisfaction without collapsing diversity.
> >
> > > **Q5: Inference-time overhead. Because CoG performs multiple partial denoising passes, its inference cost should increase relative to one-shot sampling. The paper does not report runtime or complexity analysis. What about inference-time overhead? Quantifying this would clarify practical usability. Quantify inference-time overhead and number of denoising stages.**
> >
> > Thank you for raising this question. We would like to emphasize that the additional computational cost introduced by the progressive stages does not significantly impact the overall efficiency.
> >
> > All experiments were conducted on a single NVIDIA RTX A5000 GPU. For generating 500 samples, the graph-based latent diffusion baseline requires only about 40 seconds in the one-shot setting. Each additional progressive stage in CoG adds only about 25 seconds, meaning that the extra computation overhead is just 0.05 seconds per sample. As shown in Sec. 5.3, we apply 3 denoising stages. Thus, even with multiple stages, the total inference time remains practical and well justified by the performance improvements CoG provides.

---

> > > ### Author Response · Authors · 2025-12-02
> > > **Official Comment by Author(Part 3)**
> > >
> > > > **Q6: Clarification on “Ground Truth” in Figure 3. The “ground truth” molecules are simply the dataset references paired with each prompt, not experimentally verified samples. Hence Figure 3 reflects reconstruction fidelity rather than true generalization, and this should be stated explicitly.**
> > >
> > > Sorry for the confusion. In our setting, these molecules are reference structures from the dataset that are paired with each textual description, and the figure illustrates reconstruction fidelity, which states how well each method recovers the intended structure given the corresponding prompt. We will revise the manuscript to explicitly refer to these molecules as “reference molecules” to avoid confusion.
> > >
> > > > **Q7: Please specify the exact model variant used with CoG (plain vs post-aligned GraphLDM).**
> > >
> > > Thanks for the request about model variant clarification. Specifically, we first apply the post-alignment strategy after learning the latent space of molecules: the VAE is trained until the training loss stabilizes, then segmentation alignment is performed, and the molecular encoder is subsequently frozen while only the text encoder is fine-tuned, together with a slight adjustment to the diffusion model to accommodate the aligned semantic space. This produces a robust and semantically aligned latent diffusion backbone, which we refer to simply as GraphLDM throughout the paper. The post-alignment step substantially reduces the semantic mismatch between the prompt conditioning vector and the molecular latent representation, allowing the diffusion model to resolve fine-grained structural details rather than struggle with alignment noise. We will make this explicit in Sec. 4.4 of the revised manuscript to avoid any ambiguity between the plain and post-aligned variants.
> > >
> > > > **Q8: Evaluate CoG on larger or out-of-distribution molecules to test compositional generalization.**
> > >
> > > We appreciate the reviewer’s suggestion. We address this concern through an ablation study in which the model is trained on ChEBI-20 but evaluated on the PubChemSTM subset, which is fully out-of-distribution both in molecular scale and structural diversity. The results demonstrate that CoG improves performance not only for the plain 3M-Diffusion model but also for the post-aligned GraphLDM backbone, confirming that the staged conditioning strategy generalizes beyond the training distribution. These findings indicate that CoG enhances compositional generalization even when applied to molecules structurally different from those seen during training.
> > >
> > >
> > > ### Performance on the PubChemSTM subset (models trained on ChEBI-20, in %)
> > >
> > > | **Method** | **HM** | **Q-Cov** | **Q-Nov** | **BQI** |
> > > |------------|--------|-----------|-----------|---------|
> > > | **Graph-based methods** |||||
> > > | 3M diffusion          | 92.70 | 34.34 | 8.56  | 54.21 |
> > > | 3M diffusion + CoG    | 97.61 | 36.97 | 8.69  | 54.91 |
> > > | GraphLDM              | 90.60 | 34.55 | 9.70  | 54.29 |
> > > | GraphLDM + CoG        | **98.44** | **38.79** | **10.26** | **55.66** |

---

### Official Review · Reviewer_yyYr · 2025-11-02

**Soundness:** 2
**Presentation:** 3
**Contribution:** 2
**Rating:** 4
**Confidence:** 4

**Summary:**

The paper proposes Chain-of-Generation (CoG), a training-free progressive latent diffusion framework for text-guided molecular design. Instead of using one-shot conditioning that struggles with complex prompts, CoG decomposes a natural-language description into semantically ordered sub-prompts and incrementally guides the diffusion process from coarse scaffolds to fine-grained details, enhancing controllability and interpretability. A post-alignment contrastive learning stage further strengthens the correspondence between textual and molecular latent spaces. Experiments on ChEBI-20 and PubChem show that CoG consistently improves semantic fidelity, coverage, and structure–text alignment over SMILES-based and graph-based baselines.

**Strengths:**

+ The paper introduces a novel and conceptually elegant chain-of-thought–inspired framework for progressive molecular diffusion generation.
+ The proposed method is training-free and easily integrates with existing latent diffusion models, improving interpretability and controllability.
+ Comprehensive experiments and chemically meaningful graph-based metrics demonstrate consistent performance gains over strong baselines.

**Weaknesses:**

+ Although the paper identifies three main limitations of one-shot conditioning, the proposed CoG framework does not show sufficiently strong improvements in addressing these issues.
+ The evaluation focuses mainly on structural fidelity without testing functional or physicochemical properties of generated molecules.
+ The method relies heavily on external LLM-based prompt segmentation, which may introduce instability or semantic errors.
+ Reported performance gains over GraphLDM are modest and may not fully justify the framework’s added complexity.
+ The so-called “training-free” claim is weakened by the inclusion of a contrastive post-alignment learning phase.
+ The diversity analysis is limited, lacking deeper studies on scaffold or property-level novelty.

**Questions:**

1. Could the authors quantitatively demonstrate how CoG ensures the faithful realization of each sub-goal within multi-component prompts? (Respond to C2)
2. The claim of interpretability is relatively weak. CoG does not demonstrate substantially stronger interpretability compared to prior methods, as its interpretability mainly stems from LLM-based prompt segmentation, which itself introduces additional uncertainty. I suggest that the authors conduct more experiments or analyses to support this claim directly.
3. In Tables 4–7, all methods show better performance as the number of prompt pieces increases. Does this suggest that these models, not only CoG, can already handle multi-component prompts to some extent? If so, does this weaken the claim in limitation C2 regarding missing substructures, and how does CoG’s improvement differ qualitatively from the general trend?
4. Could the authors provide stronger empirical evidence for the claim of “ambiguous conditioning” (C3)? Specifically, can they quantify ambiguity reduction or show how CoG improves structural placement and semantic disambiguation compared to one-shot baselines? It also remains unclear whether CoG still exhibits residual confusion and to what extent prior methods were actually hindered by ambiguous conditioning.

It is undeniable that overly complex prompts can negatively affect molecular design quality, and addressing this issue would greatly enhance the practical value of such methods. However, the proposed CoG framework has not been convincingly shown to alleviate this problem. The authors could provide additional experimental evidence or analyses to demonstrate CoG’s effectiveness in handling this. Strengthening this part would make the contribution more convincing.

---

> ### Author Response · Authors · 2025-12-02
> **Official Comment by Author (Part 1)**
>
> > **Q1:  Although the paper identifies three main limitations of one-shot conditioning, the proposed CoG framework does not show sufficiently strong improvements in addressing these issues, and may not fully justify the framework’s added complexity. Could the authors quantitatively demonstrate how CoG ensures the faithful realization of each sub-goal within multi-component prompts?**
>
> Thank you for raising these questions. We apologize for any misunderstanding and our goal is to demonstrate that CoG improves both (1) the proportion of samples that satisfy the basic textual requirements of the prompt and (2) the proportion that additionally satisfy the fine-grained, high-quality structural requirements, all while preserving generative creativity. To measure these two aspects in a balanced and unified way, we introduce a new metric: the fair balanced harmonic mean (HM) of max-normalized $P_{base}$​ and $P_{qual}$, as detailed in Appendix C. This metric captures the trade-off between basic semantic correctness and high-quality structural realization. Under this balanced evaluation, CoG achieves clear and significant improvements over the one-shot baseline, showing that it not only recovers more of the core semantic sub-goals but also more faithfully realizes the finer components of multi-part prompts.
>
> Regarding the concern about added complexity, we emphasize that the overhead of progressive stages is minimal. All experiments were run on a single NVIDIA RTX A5000 GPU. Generating 500 samples with the graph-based latent diffusion baseline takes about 40 seconds in the one-shot setting. Each additional CoG stage introduces about only 25 seconds (0.05 sec per sample). As shown in Sec. 5.3, we use 3 denoising stages, and even with these stages, the total inference time remains practical and well justified by the substantial fidelity improvements CoG provides. Thus, the observed performance gains outweigh the modest additional computation, and the method remains efficient, training-free, and easy to integrate into existing diffusion-based generators.
>
> ### Performance on ChEBI-20 (in %)
>
> | **Method** | **HM** | **Q-Cov** | **Q-Nov** | **Validity** | **BQI** |
> |-----------|--------|-----------|-----------|--------------|--------|
> | **SMILES-based methods** ||||| |
> | MolT5 small      | 73.62 | 24.25 | 4.96 | 78.45 | 51.45 |
> | MolT5 large      | 60.82 | 20.35 | 2.23 | 98.08 | 55.95 |
> | ChemT5 small     | 68.76 | 23.88 | 3.25 | 96.99 | 57.60 |
> | ChemT5 base      | 70.78 | 25.08 | 3.45 | 97.25 | 57.87 |
> | BioT5 base       | 60.56 | 20.88 | 4.57 | 99.98 | 59.11 |
> | BioT5 plus       | 49.29 | 15.44 | 2.47 | **100** | 56.76 |
> | LDMOL            | 12.39 | 1.79 | 0.14 | 99.34 | 52.12 |
> | **Graph-based methods** ||||| |
> | 3M diffusion         | 71.80 | 40.32 | 9.88 | **100** | 56.30 |
> | 3M diffusion + CoG   | 75.38 | 42.83 | 10.27 | **100** | 57.43 |
> | GraphLDM            | 79.74 | 43.05 | 11.02 | **100** | 59.43 |
> | GraphLDM + CoG      | **84.97** | **45.63** | **11.10** | **100** | **60.49** |
>
>
> ### Performance on PubChem (in %)
>
> | **Method** | **HM** | **Q-Cov** | **Q-Nov** | **Validity** | **BQI** |
> |-----------|--------|-----------|-----------|--------------|--------|
> | **SMILES-based methods** ||||| |
> | MolT5 small      | 48.52 | 10.88 | 1.04 | 72.93 | 44.23 |
> | MolT5 large      | 54.22 | 16.27 | 1.45 | 94.15 | 51.44 |
> | ChemT5 small     | 63.69 | 20.05 | 2.47 | 91.91 | 52.69 |
> | ChemT5 base      | 66.03 | 20.77 | 2.75 | 90.65 | 52.97 |
> | BioT5 base       | 52.93 | 16.78 | 2.89 | **100** | 54.74 |
> | BioT5 plus       | 53.24 | 17.11 | 3.08 | **100** | 55.05 |
> | LDMOL            | 40.42 | 4.37 | 0.57 | 98.21 | 53.34 |
> | **Graph-based methods** ||||| |
> | 3M diffusion         | 69.68 | 40.07 | 10.78 | **100** | 55.29 |
> | 3M diffusion + CoG   | 74.11 | 42.84 | 10.80 | **100** | 56.35 |
> | GraphLDM            | 74.54 | 40.88 | **12.56** | **100** | 57.79 |
> | GraphLDM + CoG      | **78.00** | **43.36** | 11.95 | **100** | **58.17** |
>
>
> > **Q2: The evaluation focuses mainly on structural fidelity without testing functional or physicochemical properties of generated molecules.**
>
> We agree that such analyses are valuable for downstream drug-discovery applications. However, our paper is scoped specifically around semantic–structural fidelity, which is how accurately the model transforms the structural information implied by the text prompt into molecular motifs and substructures. We believe that structural fidelity is a prerequisite for drug discovery. We note that the community’s current focus is primarily on advancing structural generation, as molecular structure naturally encodes functional and physicochemical properties. Strengthening structural fidelity is therefore a foundational step. Once structural alignment is robust, future extensions to property-level or functional assessments will become feasible.

---

> > ### Author Response · Authors · 2025-12-02
> > **Official Comment by Author (Part 2)**
> >
> > > **Q3: The method relies heavily on external LLM-based prompt segmentation, which may introduce instability or semantic errors.
> > We appreciate the reviewer’s concern regarding potential instability from LLM-based prompt segmentation. To rigorously test reliability, we first manually segment all prompts according to human-interpretable chemical semantics and then compare these with the LLM-generated segments. Our segmentation follows a consistent rule set, ordered from heavy to light elements of chemical structure: (1) main structural core or primary ring system, (2) medium-sized functional groups or secondary ring systems, and (3) small atomic or functional-group modifiers.**
> >
> > A segmentation is considered successful if the LLM correctly identifies these components and ranks them in the appropriate coarse-to-fine order. We perform this evaluation three independent times, and the LLM achieves an average success rate of 99% ± 1%, demonstrating both stability and semantic consistency. This verification ensures that the segmentations used in CoG reflect human-valid chemical structure decomposition.
> >
> >
> > >**Q4: The so-called “training-free” claim is weakened by the inclusion of a contrastive post-alignment learning phase.**
> >
> > We apologize for any confusion. Our paper presents two independent contributions, which are not dependent on each other. First, CoG is a fully training-free inference strategy: the staged prompting and sampling mechanism can be applied directly to any diffusion-based generative model without modifying its architecture or performing additional training. Second, the contrastive post-alignment phase is a separate optional enhancement used only to establish a stronger latent diffusion backbone (GraphLDM). It is not part of CoG.
> >
> >
> > To make this distinction explicit, we include an ablation where CoG is applied directly to 3M-Diffusion without any post-alignment, and the results demonstrate that CoG still provides substantial and consistent improvements. This confirms that the benefits attributed to CoG arise solely from its progressive conditioning strategy, and that the “training-free” claim fully holds independent of the post-alignment module. Please refer to the table above in Q1.
> >
> >
> > > **Q5: The claim of interpretability is relatively weak. CoG does not demonstrate substantially stronger interpretability compared to prior methods, as its interpretability mainly stems from LLM-based prompt segmentation.**
> >
> > Thanks for raising this concern. In our framework, besides the explicit semantic segmentation of the prompt, interpretability also arises precisely from the corresponding coarse-to-fine conditioning schedule. Unlike prior methods, which inject all semantic information at once and thus offer no insight into how individual components influence the generative trajectory, CoG makes the contribution of each sub-goal observable and controllable at different stages of denoising. Our ablation study in Table 2 directly supports this: coarse-to-fine ordering improves structural fidelity, whereas reversing the order systematically degrades performance, demonstrating that the staged semantic conditioning has interpretable and measurable effects on the generation process.
> >
> > > **Q6, In Tables 4–7, all methods show better performance as the number of prompt pieces increases. Does this suggest that these models, not only CoG, can already handle multi-component prompts to some extent? If so, does this weaken the claim in limitation C2 regarding missing substructures, and how does CoG’s improvement differ qualitatively from the general trend?**
> >
> > Thanks for this insightful question. CoG can take better information on all subsets. The trend observed in Tables 4-7, where all methods perform better as the number of prompt pieces increases, is expected and does not contradict our limitation C2. As more pieces of the prompt are provided, the instruction becomes progressively clearer and more informative, naturally giving any model a stronger guiding signal. Consequently, even baseline models can achieve higher structural fidelity simply because they receive more explicit structural cues.
> >
> >
> > However, this general trend does not weaken our claim. Baseline models still struggle when prompting multiple searching objectives simultaneously, showing inconsistencies in recovering omitted substructures. In contrast, CoG consistently provides improvements across all prompt subsets. This demonstrates that CoG’s benefit rides on the increased clarity of both short and long prompts; it enhances semantic completeness and structural consistency.

---

> > > ### Author Response · Authors · 2025-12-02
> > > **Official Comment by Author(Part3)**
> > >
> > > > **Q7, Could the authors provide stronger empirical evidence for the claim of “ambiguous conditioning” (C3)? Specifically, can they quantify ambiguity reduction or show how CoG improves structural placement and semantic disambiguation compared to one-shot baselines? It also remains unclear whether CoG still exhibits residual confusion and to what extent prior methods were actually hindered by ambiguous conditioning.**
> > >
> > > Thank you for this insightful question. In our setting, ambiguity arises when a single, one-shot conditioning vector is required to simultaneously encode multiple, potentially competing sub-goals in a complex prompt. Under this regime, the model must search in the latent space for a solution that satisfies all constraints at once, which often leads to missing substructures as shown in Fig.2. We will clarify this in the revised manuscript.
> > >
> > > CoG addresses this by decoupling complex prompts into a multi-stage curriculum of sub-objectives. Instead of forcing the model to resolve all semantic components in a single shot, CoG first conditions on coarse aspects and then gradually adds finer-grained substructures. This progressive schedule drives the generative process in a finer, more controllable manner and reduces the competition between different semantic targets.

---

### Meta-Review · Area_Chair_2dFF · 2026-01-05

**Summary:**

Reviewer yyYr found that the empirical evidence does not convincingly support the main claims. Improvements over strong baselines are modest, and it is unclear whether CoG meaningfully alleviates the identified limitations of one-shot conditioning. The evaluation focuses mainly on structural metrics, with limited analysis of functional properties, diversity, or scaffold-level novelty. Claims of improved interpretability are weak and largely rely on LLM-based prompt segmentation, which introduces additional uncertainty, and the inclusion of a post-alignment learning phase weakens the “training-free” claim.

Reviewer TvqL raised concerns, including limited novelty scope, unclear attribution of gains, lack of quantitative faithfulness metrics, controllability-novelty trade-off not analyzed, missing practical evaluation, and ambiguity in “ground truth” usage.

Reviewer 4j7r raised major concerns, including limited contribution, weakly supported core assumption, unclear impact on validity, insufficient experimental scope and comparisons, and questionable evaluation framing.

**Reviewer Concerns:**

Reviewer yyYr: The rebuttal effectively addresses several major technical and clarity concerns, particularly around sub-goal realization, training-free claims, segmentation stability, and interpretability. However, questions regarding evaluation breadth, ambiguity quantification, and the overall strength of evidence for CoG’s practical advantage remain partially unresolved.

Reviewer TvqL: The rebuttal addresses most of the major concerns. The authors clearly clarify the scope of novelty, positioning CoG as a training-free inference strategy rather than a new architecture. The training vs. inference attribution concern is convincingly resolved through new ablations showing consistent gains from CoG on a plain diffusion model. The authors also quantify inference-time overhead, showing it to be modest, clarify the use of reference molecules instead of true ground truth, and provide out-of-distribution evaluations demonstrating improved compositional generalization. Some concerns remain partially outstanding. While the authors justify why existing tools such as SMARTS or Text2Mol are not directly applicable, the lack of direct, interpretable prompt-faithfulness metrics beyond the proposed harmonic-mean score remains a limitation. In addition, the controllability-novelty trade-off is explained but not deeply analyzed empirically. Overall, the rebuttal substantially strengthens the paper, though quantitative semantic evaluation remains an open issue.

Reviewer 4j7r: The rebuttal successfully addresses the core technical misunderstandings, particularly regarding attribution of gains, the training-free nature of CoG, and the coarse-to-fine assumption. However, broader generality, experimental coverage, and the practical impact of the improvements remain only partially resolved. As a result, while the rebuttal strengthens the paper’s technical clarity, it may not fully overcome concerns about scope and significance.

**Reviewer Scores:**

Based on the rebuttal, Reviewer yyYr would likely increase the score slightly, from 4 to 6 (marginally above the acceptance threshold).

Given the strength of the rebuttal, the AC expect the reviewer would increase the score from 4 to 6 (marginally above the acceptance threshold).

Based on the substance of the rebuttal and the nature of the original concerns, the AC would expect Reviewer 4j7r to increase the score modestly, but not enough to cross the acceptance threshold.

Overall, the AC considers the paper to be borderline after the rebuttal. While the rebuttal addresses several concerns, a number of key issues remain only partially resolved. As a result, the AC recommends rejection at this time and encourages the authors to consider resubmission to a future venue after further strengthening the work.

---

### Decision · Program_Chairs · 2026-01-26

Reject